# Differences in reward biased spatial representations in the lateral septum and hippocampus

Hannah S Wirtshafter[1,2]*, Matthew A Wilson[1,2,3]

[1]Department of Biology, MassachusettsInstitute of Technology, Cambridge, United States; [2]Picower Institute for Learning andMemory, Massachusetts Institute of Technology, Cambridge, United States; [3]Department of Brain and Cognitive Sciences, Massachusetts Institute of Technology, Cambridge, United States

**Abstract** The lateral septum (LS), which is innervated by the hippocampus, is known to represent spatial information. However, the details of place representation in the LS, and whether this place information is combined with reward signaling, remains unknown. We simultaneously recorded from rat CA1 and caudodorsal lateral septum in rat during a rewarded navigation task and compared spatial firing in the two areas. While LS place cells are less numerous than in hippocampus, they are similar to the hippocampus in field size and number of fields per cell, but with field shape and center distributions that are more skewed toward reward. Spike cross-correlations between the hippocampus and LS are greatest for cells that have reward-proximate place fields, suggesting a role for the LS in relaying task-relevant hippocampal spatial information to downstream areas, such as the VTA.

## Introduction

The lateral septum (LS), which is innervated by all CA fields of the hippocampus, contains place cells, and processes spatial information received from the hippocampus (*Bezzi, 2005*; *Olton et al., 1978*; *Risold and Swanson, 1997*; *Wirtshafter and Wilson, 2019*; *Zhou et al., 1999*). It has been hypothesized that this spatial information is sent to downstream areas, such as the hypothalamus and ventral tegmental area (VTA) for use in reinforcement and reward seeking (*Jiang et al., 2018*; *Luo et al., 2011*; *Swanson et al., 1981*; *Sweeney and Yang, 2015*). The LS has also been implicated in contextual reward seeking and the formation of conditioned place preferences (*Jiang et al., 2018*; *Luo et al., 2011*). However, little is known about the septal representation of space during reward seeking and what, if any, transformation place information undergoes from its representation in the hippocampus to the lateral septum remains unknown.

The CA fields of the hippocampus contain place cells that show preferential firing when an animal traverses a specific location, the 'place field' (*O'Keefe and Dostrovsky, 1971*; *O'Keefe and Nadel, 1978*). Extensive work has been done characterizing these fields, including their size (*Fenton et al., 2008*; *Lyttle et al., 2013*), skew (*Mehta et al., 2000*), dependence on experience (*Frank et al., 2004*; *Lee et al., 2004*; *Mehta et al., 1997*; *Mehta et al., 2000*; *Skaggs and McNaughton, 1998*), responsiveness to environment and context (*Hasselmo and Eichenbaum, 2005*; *Knierim, 2002*; *Knierim and Hamilton, 2011*), and their distributions around goal locations (*Dupret et al., 2010*; *Hollup et al., 2001*; *Kobayashi et al., 1997*; *Kobayashi et al., 2003*; *Lee et al., 2006*). Spatially specific firing has also been well documented and characterized in other brain areas, including in the entorhinal cortex (*Fyhn et al., 2004*; *Hafting et al., 2005*; *Quirk et al., 1992*), where this firing is believed to play a role in path integration (*Fyhn et al., 2007*; *Knierim et al., 2014*; *Monaco et al., 2011*; *Moser et al., 2008*), and in the lateral septum (*Bezzi et al., 2002*; *Kita et al., 1995*;

*For correspondence:
hsw@mit.edu

Competing interests: The authors declare that no competing interests exist.

*Leutgeb and Mizumori, 2002*; *Monaco et al., 2019*; *Takamura et al., 2006*; *Wirtshafter and Wilson, 2019*; *Zhou et al., 1999*), though much less is known about the potential role of LS place fields in spatial navigation.

Given the anatomical location of the LS between areas involved in spatial navigation such as the hippocampus, and reward/reinforcement related areas such as the VTA (*Luo et al., 2011*; *Risold and Swanson, 1997*), and the established role of the LS in reinforcement seeking (*Jiang et al., 2018*; *Mathieu-Kia et al., 1998*; *McGlinchey and Aston-Jones, 2018*; *Olds and Milner, 1954*; *Oshima and Katayama, 2010*; *Sotomayor et al., 2005*), we hypothesized that LS place fields would be preferentially located at reward locations to a greater extent than hippocampal place fields and contain more reward related information.

In the following experiment, we simultaneously recorded cells from the hippocampal CA1 field and the caudodorsal lateral septum during a spatial navigation task. The caudodorsal LS was chosen as it is the area of the LS most heavily innervated by the hippocampus and known to send the most projections to the VTA (*Luo et al., 2011*; *Risold and Swanson, 1997*). We characterized lateral septal place field firing and compared it to that seen in CA1. We found that, although not as common as in the hippocampus, the lateral septum contains a large number of place cells with fields that have properties similar to the place cells of the hippocampus, including similar field size and similar number of fields per cell. However, in contrast to HPC place cells, LS cells are slightly less accurately spatially tuned. Additionally, within a firing field, LS cells ramped up their firing in the direction of reward, and place fields in the lateral septum were preferentially biased toward reward locations compared to the hippocampus. We found that HPC and LS cells with reward proximate fields had significantly higher correlated activity than cells with fields located further from reward. We suggest that reward-related spatial information from the hippocampus is preferentially represented in the LS, and we provide three models by which this could occur. We suggest that this relayed information is used downstream to direct the animal to significant or rewarded locations.

## Results

In order to characterize the role of the LS in reward-driven spatial navigation, we recorded the activity of 452 caudodorsal LS units and 178 CA1 hippocampus units in six male Long Evans rats (*Figure 1A–B*). We specifically targeted the caudodorsal LS as it receives the a large innervation from the hippocampus and is the primary source of LS projections to the VTA (*Luo et al., 2011*; *Risold and Swanson, 1997*). Units were recorded when the animals performed a spatial working memory task on a double-sided T maze, used previously to characterize spatial activity in multiple regions, including in the hippocampus (*Gomperts et al., 2015*; *Jones and Wilson, 2005a*; *Jones and Wilson, 2005b*; *Siegle and Wilson, 2014*), prefrontal cortex (*Jones and Wilson, 2005a*; *Jones and Wilson, 2005b*) VTA (*Gomperts et al., 2015*), and lateral septum (*Wirtshafter and Wilson, 2019*). In this task, animals are forced to a randomly chosen side of one of the Ts (the forced arm), and must run down a center stem and choose the same side of the opposite T (the choice arm) to be rewarded (*Figure 1C–D*). The structure of this task results in mirrored turns and track traversals which help control for behavioral variability, allowing us to look specifically as the effect of reward approach without confounding variables such as turn approach.

### Place fields are less abundant in the LS as compared to the hippocampus CA1

We first sought to determine the prevalence of place fields in the cdLS, as estimates have varied wildly from one study reporting no fields (*Tingley and Buzsáki, 2018*), to estimates of about a third to half of all LS cells (*Takamura et al., 2006*; *Zhou et al., 1999*). In all recorded LS cells and CA1 principle cells, we evaluated the spatial information content in cell firing using an information measure of bits per spike. We defined a cell with place information to be a cell with a bits per spike cutoff of 0.8 bits/spike, as this cutoff has been used previously for determining spatial firing of non-hippocampal cells (*Ji and Wilson, 2007*; *Markus et al., 1994*). We did not include cells with a mean spiking rate of less than 0.05 hz, and also eliminated cells during trials where the animal did not cover the entire track at a speed of at least 12 cm/s. We found that 75.2% (124/165) of CA1 cells (*Figure 2A*) and 33.6% (127/378) of LS cells (*Figure 2B*) met or exceeded the 0.8bits/spike threshold, with the average bits/spike of a CA1 cell significantly higher than the average bits/spike of an

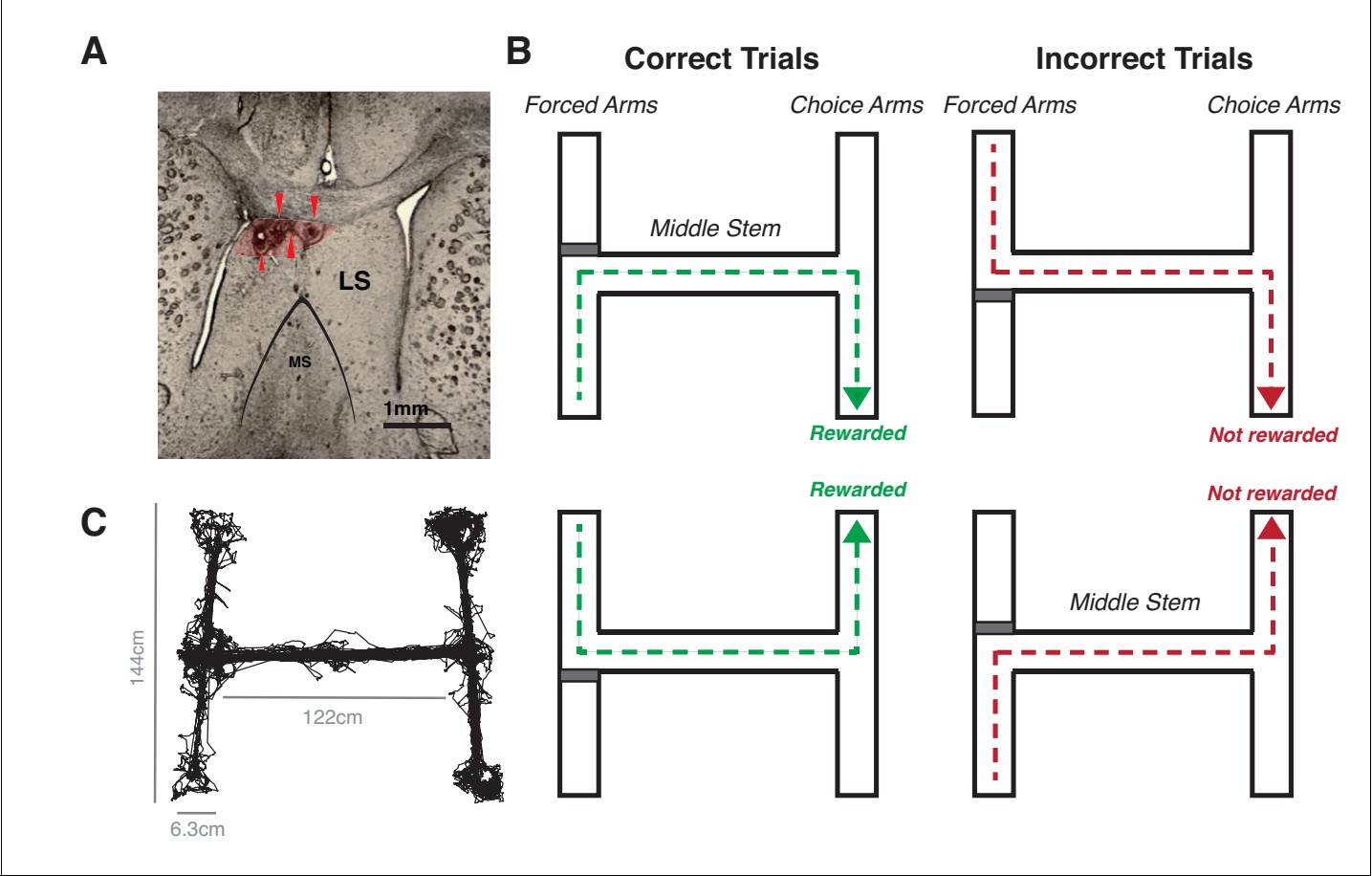

**Figure 1.** Rats were implanted with a tetrode array in the HPC and LS and run on a double sided T maze. (**A**) Brain section from implanted rat, showing the lateral septum and electrolytic lesions made after recording. The red arrows mark lesions at the tetrode tips. Red shaded area indicates area where about 95% of lesions were seen. (**B**) Illustration of the maze task, which consisted of two phases: in the forced choice phase, animals were randomly forced with a block (represented by a red rectangle in a schematic) to either side of the T. In the free choice phase, animals had to choose, at the opposite end of the maze, the same side to which they were forced. If they made the correct choice, animals were rewarded with a sucrose and chocolate mixture. (**C**) Tracked position during one 30 min session.

LS cell (HPC mean 1.54+−1.2 bits/spike, LS mean 0.73+−0.7 bits/spike, two-tailed two sample t-test t(541)=9.42, p<0.001) (*Figure 2C*) (We also computed mutual information, see *Figure 2—figure supplement 1*). To ensure that the representation of space was different than would be expected from random Poisson firing, we created 454 artificial LS units using Poisson firing and the mean firing rates of the recorded LS units (*Figure 2B*, inset). The distribution of the bits/spike for the artificial units was highly significantly different than the distribution of bits/spike for actual units (KS test, p<$10^{-15}$), and only 7.96% of artificial units had bits/spike measurements of greater than 0.8. The average bits/spike for the artificial units was 0.43, compared to an average value of 0.73 for actual units (two-tailed two sample t-test, t(818)=6.72, p<$10^{-10}$).

Because bits/spike is sensitive to differences in firing rate, we also calculated bits per second for the CA1 and LS units. We found a mean of 1.34+−1.4bits/sec for CA1 cells, and 0.82+−1.1bits/sec for LS cells (*Figure 2D–E*). Units in the hippocampus had a significantly greater mean bits/sec than in the LS (two-tailed two sample t-test t(541)=4.57, p<0.001) (*Figure 2F*).

We next determined how many cells with bits/spike greater than 0.8 had definable place fields. Place fields boundaries have previously been defined in a multitude of ways, including with a flat firing rate threshold (*Rich et al., 2014*). Because firing rate for LS place fields may not follow the same criteria as hippocampal place fields and may be contingent on the LS's innate firing properties, we opted to use a threshold derived from the units' average firing rate. We identified a place field as connected area at least 15 cm long with a peak firing rate of at least two standard deviations above

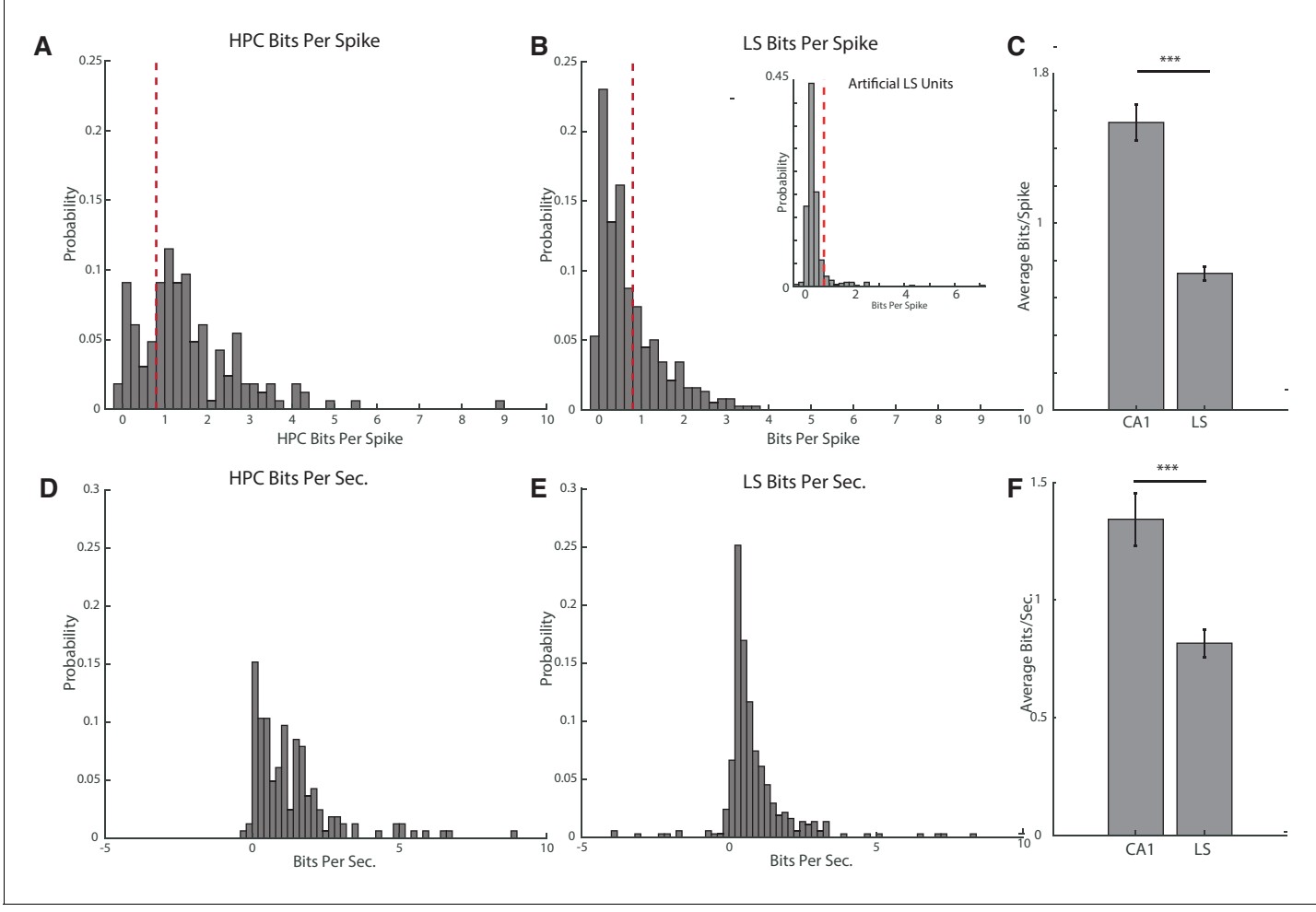

**Figure 2.** Hippocampal units have a higher average bits per spike and bits per second than septal units. (A) Bits per spike of all recorded hippocampal units with an average spike rate >0.05 hz and full track coverage. A bits per spike cutoff of 0.8bits/spike is marked with a dotted line. (B) Larger graph: same as A for units in the lateral septum. Inset: bits per spike for artificially created LS units. The distribution of the bits/spike for the artificial units was highly significantly different than the distribution of bits/spike for actual units (KS test, $p<10^{-15}$), and only 7.96% of artificial units had bits/spike measurements of greater than 0.8. The average bits/spike for the artificial units was 0.43, compared to an average value of 0.73 for actual units (two-tailed two sample t-test, t(818)=6.72, $p<10^{-10}$). (C) Comparison of bits per spike for hippocampal and septal units. The average bits/spike of a CA1 cell is significantly higher than the average bits/spike of an LS cell. HPC mean 1.54+−1.2 bits/spike, LS mean 0.73+−0.7 bits/spike, two-tailed two sample t-test t(541)=9.42, p<0.001. Error bars represent standard error. (D) Bits per second of all recorded hippocampal units with an average spike rate >0.05 hz and full track coverage. (E) Same as D for units in the lateral septum. (F) Comparison of bits per second for hippocampal and septal units. CA1 units had a mean of 1.34+−1.4bits/sec, and LS cells had a mean 0.82+−1.1bits/sec. Units in the hippocampus had a significantly greater mean bits/sec than in the LS (two-tailed two sample t-test t(541)=4.57, p<0.001). Error bars represent standard error.

The online version of this article includes the following figure supplement(s) for figure 2:

**Figure supplement 1.** Mutual information in HPC and LS cells.

the unit's mean firing rate, and the boundaries of the place field to be when firing drops below one standard deviation above mean firing rate (see Methods).

Of the 164 HPC units with bits/spike greater than or equal to 0.8, 104 units had at least one place field, with 63 of these cells with one field, 33 with two fields, seven with three fields, and a single cell with four fields (*Figure 3A,C*). Of the 127 LS cells with bits/spike greater than 0.8, 100 had at least one place field, with 54 with one field, 33 with two fields, six with three fields, five with four fields, and two with five fields (*Figure 3B,D*). The distributions of field numbers were not significantly different between CA1 and LS cells (two sample Kolmogorov-Smirnov (KS) test, p>0.5). We have previously shown that LS cells are modulated by speed and acceleration (*Wirtshafter and Wilson, 2019*). To determine the potential contribution of spatially biased speed and acceleration to our place field

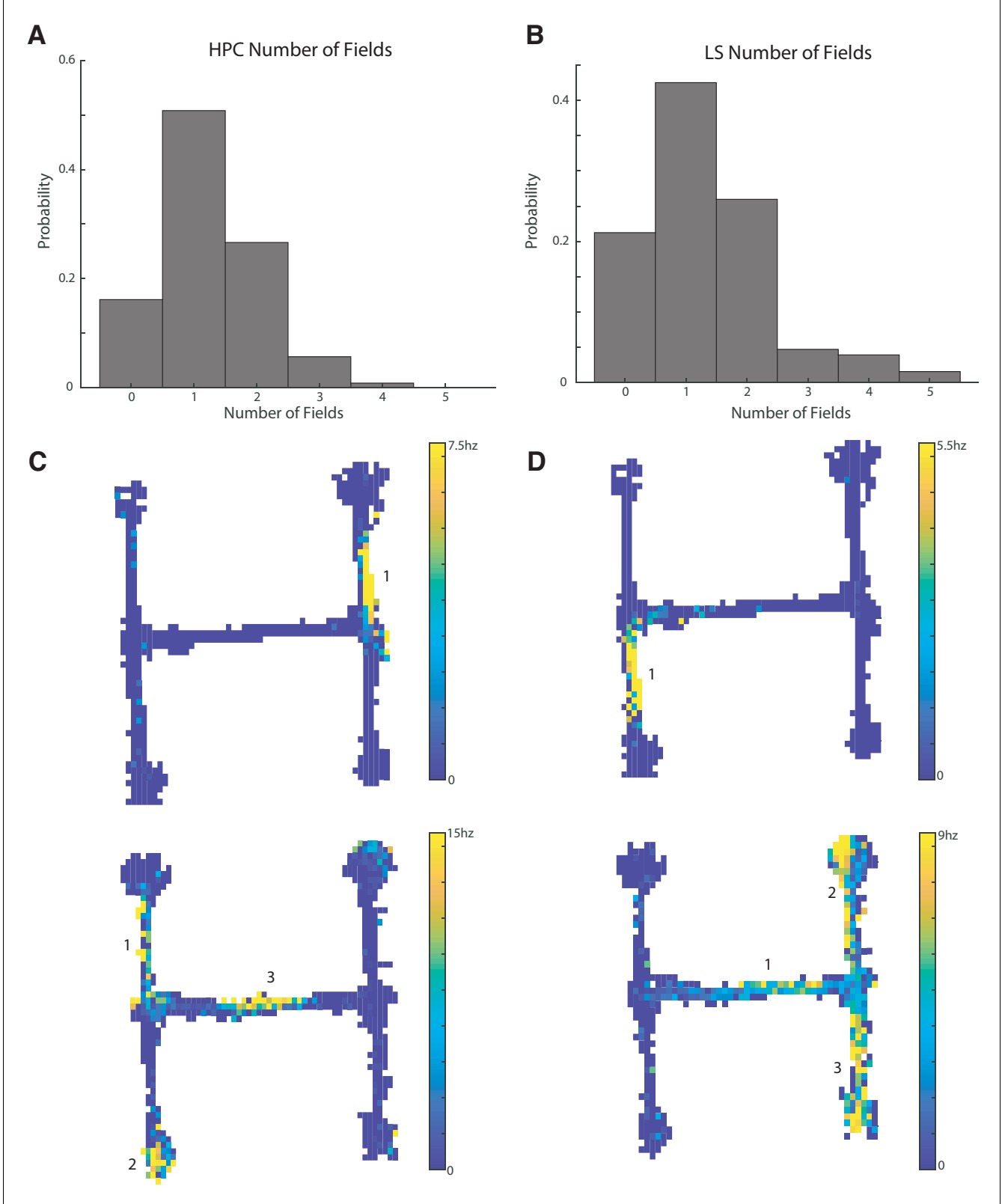

**Figure 3.** CA1 and LS place cells have a comparable number of place fields. The distribution of field numbers was not significantly different between HPC and CA1 cells (two sample Kolmogorov-Smirnov (KS) test, p>0.5). (**A**) Number of fields in CA1 cells with a bits per spike greater than or equal to 0.8 bits/spike. (**B**) Same as A for lateral septum. (**C**) Example units in the CA1. Top: Example of a unit with a single place field. Bottom: example of a unit with three place fields. (**D**) Same as C but for the LS.

*Figure 3 continued on next page*

*Figure 3 continued*

The online version of this article includes the following figure supplement(s) for figure 3:

**Figure supplement 1.** R2 values for linear regression of LS firing rate against speed and acceleration.
**Figure supplement 2.** Place field representations and characteristics.

results, we performed a multiple linear regression of each spatial bin's firing rate against the bin's average speed and acceleration. Only two cells with place fields had an r2 of >0.5, with the median $r^2$ value of 0.06 (*Figure 3—figure supplement 1*), so very little spatial firing could be explained by correlations with speed or acceleration.

In summary, out of 165 recorded CA1 cells that met spike rate and track coverage criteria, 104 (63.0%) met or exceeded a bits per spike cut off of 0.8 and had at least one place field that met the described criteria, while out of 378 recorded LS cells meeting spike rate and track coverage criteria, 100 (26.5%) met the same criteria (see *Figure 3—figure supplement 2* for additional place cell examples and representations of firing rates). Place cells in the CA1 are therefore significantly more abundant than in the cdLS (two-tailed two sample t-test t(541)=8.6, p<0.001).

## CA1 and LS place fields have comparable average sizes and are shaped by experience

We next determined if there were differences in size between hippocampal and LS place fields. In order to qualify as a place field, we applied a minimum length standard of at least 15 cm. The average length of a CA1 place field was 29.1+−14.7 cm (*Figure 4A,C*), while the average length of an LS place field was 28.8+−16.6 cm (*Figure 4B,D*). The average field lengths were not significantly different (two-tailed two sample t-test t(320)=-0.16, p>0.05).

Previous work has demonstrated that hippocampal place fields stabilize and become more tuned to position with experience (*Mehta et al., 1997*). We compared the time periods over which HPC and LS place fields become stable. For each pass through a place field, we determined how far the center (determined by maximum spiking) of the place field was from the average place field center. Both the hippocampus (*Figure 4E*) and LS (*Figure 4F*) had highly accurate place fields starting with the first pass of the place field, though, on average the HPC is slightly more accurate on the first lap as well as across all laps. (On the first lap, HPC has a mean distance of 2.7 cm versus 4.9 cm for the LS, two-tailed two sample t-test t(319)=-3.1, p<0.005. Across all laps, mean distance of 4.70 cm for the HPC versus 5.12 cm for the LS, two-tailed two sample t-test t(6792) = −2.5, p<0.05. The slight but significant decrease in accuracy from the first to later laps in the HPC can be explained by a slight shift of the place field peak toward the direction of travel (*Figure 4G*). Fields in both the HPC and the LS significantly increase firing in their place field with experience on the track (*Figure 4G–H*).

## CA1 and LS place fields have different skew and location distributions

We also wondered if bias toward reward location could also be seen within a place field. To determine this, looked at place field skew as a function of travel direction, computing skew for a place field using total within-field firing (for skew during individual laps, see *Figure 5—figure supplement 1*). We found that, when travelling toward a reward site, cells in the hippocampus were skewed positively toward the direction of travel, while, when traveling away from a reward site, cells were skewed negatively away from the direction of travel, and this difference was significant (two-tailed two sample t-test t(207)=-2.1, p<0.05) (*Figure 5A,B*, see also *Figure 5—figure supplement 2–3*). Conversely, when traveling toward a reward site, LS place fields were skewed negatively away from the direction of travel, and when traveling away from a reward site, the fields were skewed positively toward the direction of travel, though this difference is not significant (two-tailed two sample t-test t(246)=-1.7, p=0.09) (*Figure 5A,C*). Although the HPC and LS both contains uni- and bi- directional place fields, there was no significant difference for values of skew based on whether a place cell was uni or bi directional (both two-tailed two sample t-tests, p>0.05, see *Figure 5—figure supplement 4*). Because skew can be sensitive to changes in speed, we also computed the firing rate asymmetry index (FRAI, see methods) (*Mehta et al., 2000*) for HPC and LS cells, and found a highly

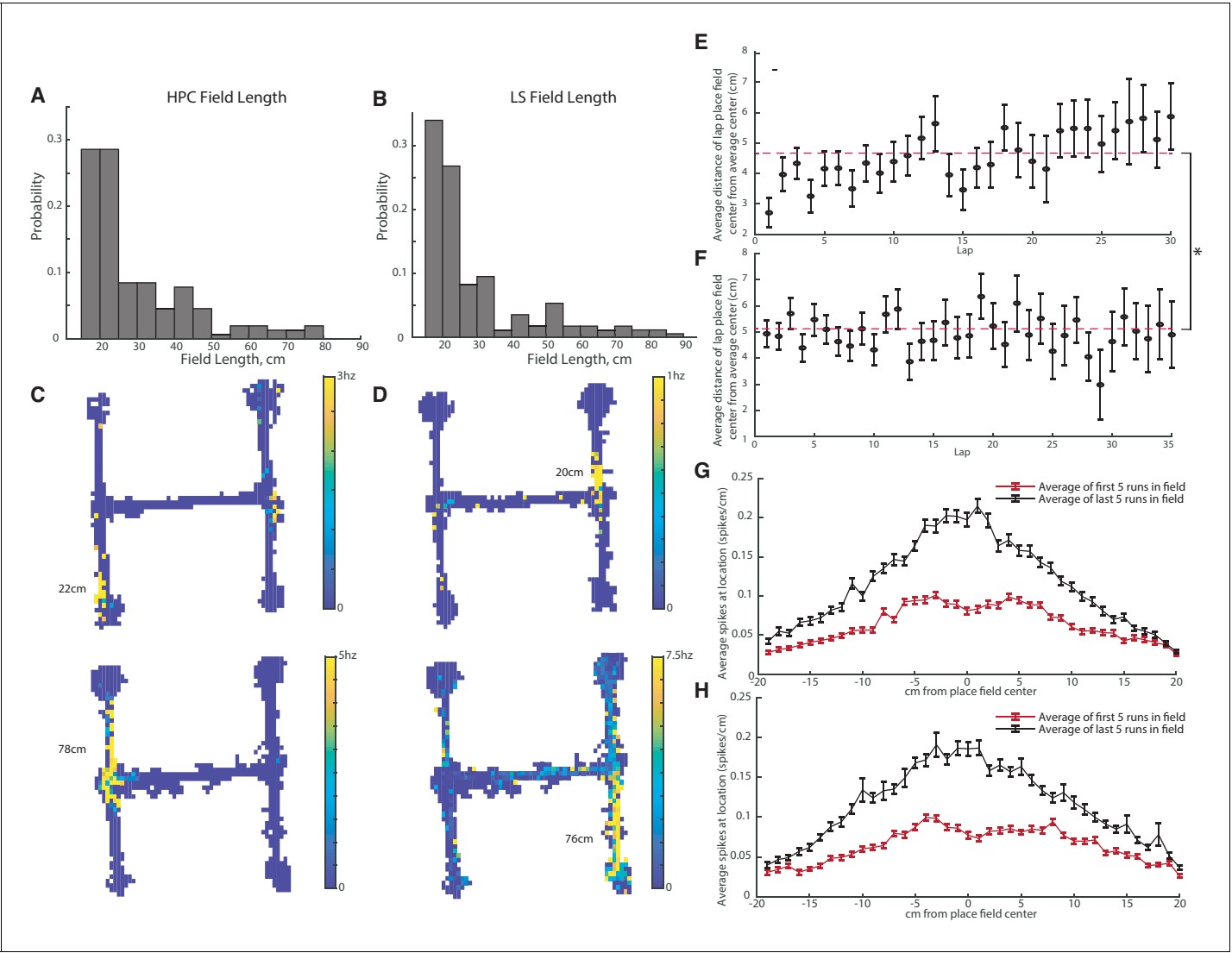

**Figure 4.** CA1 and LS place fields have comparable lengths and are modified with experience. The average field lengths in CA and LS were not significantly different (two-tailed two sample t-test t(320)=-0.16, p>0.05). (A) Distribution of field lengths in the CA1. The average length of a CA1 place field was 29.1+-14.7cm. (B) Distribution of field lengths in the LS. The average length of an LS place field was 28.8+-16.6cm. (C) Example fields in the CA1. Top: example of a shorter field with a length of 22cm. Bottom: example of a longer field with a length of 78cm. (D) Example fields in the LS. Top: example of a shorter field with a length of 20cm. Bottom: example of a longer field with a length of 76cm. (E) Distance of highest HPC firing location from mean place field center during each lap. Average distance across all laps is marked with a dotted line The average distance from the field center in lap one is slightly but significantly different than the average field distance from the center in lap 30 (two-tailed two sample t-test t(177)=-2.7, p<0.01). (F) Distance of highest LS firing location from mean place field center during each lap. The average distance from the field center in lap one was not significantly different than the average field distance from the center in lap 30 (two-tailed two sample t-test t(207)=0.2, p>0.05). On average, compared to the LS, the HPC is slightly more accurate on the first lap as well as across all laps. (On the first lap, HPC has a mean distance of 2.7cm versus 4.9cm for the LS, two-tailed two sample t-test t(319)=-3.1, p<0.005. Across all laps, mean distance of 4.70cm for the HPC versus 5.12cm for the LS, two-tailed two sample t-test t(6792)=-2.5, p<0.05). (G) HPC spiking frequency per cm as a function of location around the place field center. The average spiking rate/cm of the first five runs through the field is marked in red, and the last five runs in black. Error bars represent standard error. The difference between the first and last run averages was significant for all cm values except +18cm and +20cm (all two-tailed two sample t-test). (H) Same as G but for the LS. The difference between the first and last run averages was significant for all cm values except -18cm (all two-tailed two sample t-test).

significant relationship between skew and FRAI values (*Figure 5—figure supplement 5*), showing that skew cannot be explained by the animal's speed.

We then examined the effect of reward proximity on skew for place cells that were in the immediate reward proximity (in the rewarded arms, *Figure 5D*). Comparing the mean skew values in immediate reward proximity (*Figure 5E*, see also *Figure 5—figure supplements 2–3*), skew values were

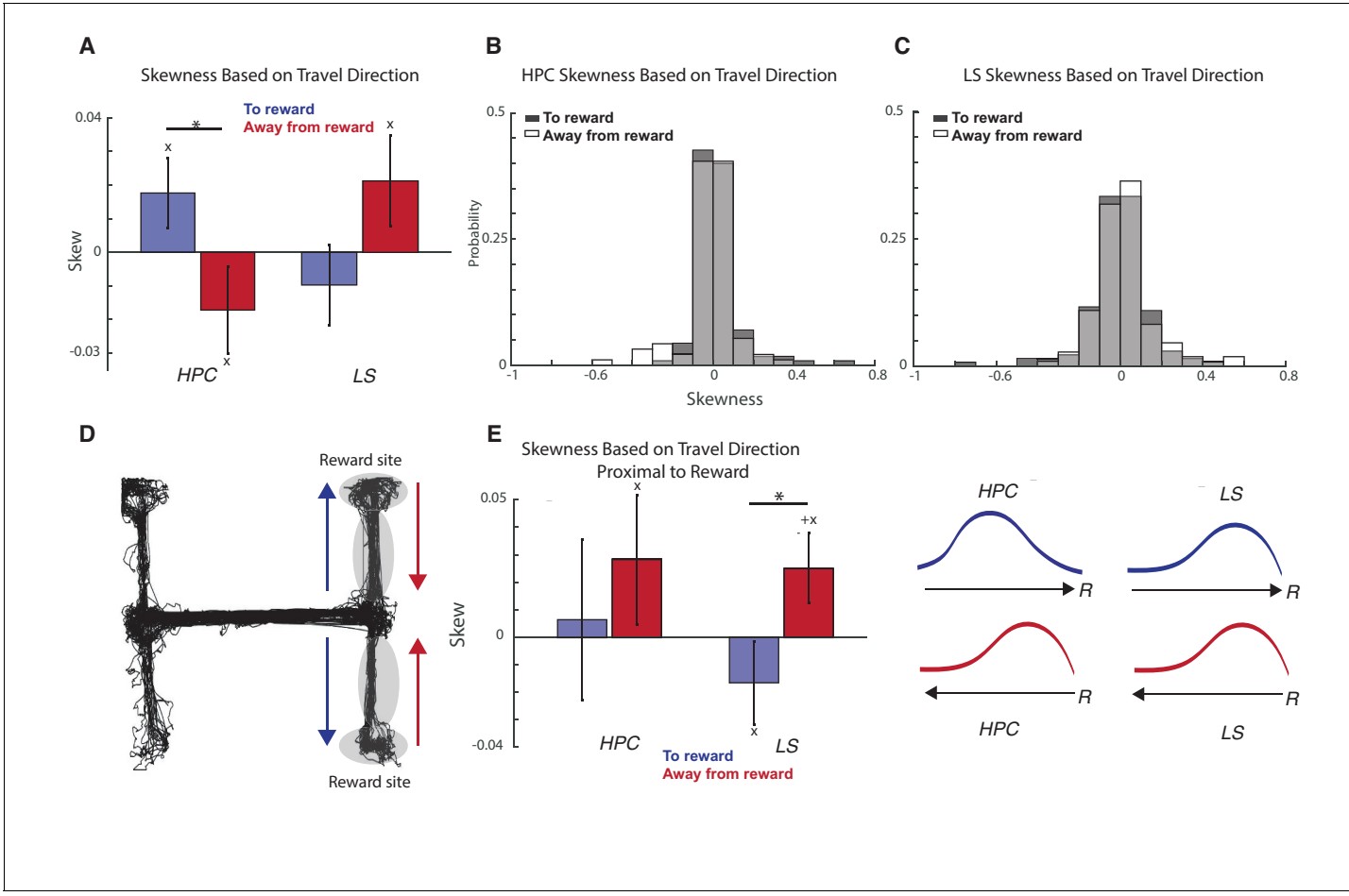

**Figure 5.** HPC and LS place cells have opposite directions of skew based on travel direction. (A) Comparison of mean skew values within immediate reward proximity. Stars indicate significant differences between multiple populations. Plus signs indicate the mean is significantly different than 0. Xs indicate significant differences from a shuffled distribution, see *Figure 5—figure supplement 3*. Error bars represent standard error. In the hippocampus, there is a significant difference in mean skews based on direction of travel (two-tailed two sample t-test t(207)=-2.1, p<0.05). (B) Graph of distributions of skews in the hippocampus based on direction of travel. (C) Graph of distributions of skews in the septum based on direction of travel. (D) Schematic showing arms of the maze examined during immediate reward proximity. Toward reward direction is marked with a blue arrow, away from reward with a red arrow. (E) Comparison of mean skew values within immediate reward proximity. Stars indicate significant differences between multiple populations. Plus signs indicate the mean is significantly different than 0. Xs indicate significant differences from a shuffled distribution, see *Figure 5—figure supplement 3*. Left: Error bars represent standard error. Skew values were not significantly different for hippocampus traveling to and from reward (two sample two sided t-test, t(74)=-0.59 p>0.05), but were significantly different for LS cells to and from reward (two sample two sided t-test, t(108)=-2.10, p<0.05). (Skew for HPC away from reward is significantly different from a shuffled sample, two sample two sided t-test, t(301)=-2.05, p<0.05. Skew for LS away from reward is significantly different from zero, one sample two sided t-test, t(58)=-2.0, p=0.05). Right: Schematic for clarification of skew relative to reward location for both directions of travel. Arrow represents direction of travel, 'R' represents reward location. Note that while HPC skew away from reward for reward proximal cells appears to have a different direction than for all HPC cells when traveling away from reward, the two means are not significantly different (two sample two sided t-test, t(131)=1.62, p>0.05).

The online version of this article includes the following figure supplement(s) for figure 5:

**Figure supplement 1.** Distribution of skew values for each pass through a field.
**Figure supplement 2.** Skew versus distance to reward site across the whole track.
**Figure supplement 3.** Distribution of skews for shuffled data.
**Figure supplement 4.** Skew is not impacted by uni- or bi- directional place fields.
**Figure supplement 5.** Skew and FRAI are highly linearly correlated in HPC and LS cells.

not significantly different for hippocampus traveling to and from reward (two sample two sided t-test, t(74)=-0.59 p>0.05), but were significantly different for LS cells to and from reward (two sample two sided t-test, t(108)=-2.10, p<0.05).

We then looked at field location for HPC and LS place fields to determine which arms of the maze were most represented in the hippocampus and LS (*Figure 6A*). Place field locations were based on maximum spiking (field center) in the place field. In the hippocampus, 49/154 of place fields occurred on the forced sides, 36/154 on the center stem, and 45/154 on the end of the choice sides (*Figure 6B,D*). (Force and choice points were excluded as they were difficult to assign to an arm, but 24 place fields occurred at the forced and choice point. For data further subdivided by location see *Figure 6—figure supplement 1*.) In the LS, 37/168 of place fields occurred on the forced sides, 40/168 on the center stem, and 69/168 on the end of the choice sides (22 occurred at choice points) (*Figure 6C,E*). The distribution of place fields is significantly different between the hippocampus and the lateral septum (Pearson's chi$^2$ test, $X^2$ = 6.03, p<0.05). The LS has about 1.4 times, proportionally, significantly more place fields in the choice side than the HPC (41.1% of total fields in the LS, versus 29.2% of total fields in the HPC, two-tailed two sample t-test t(320)=2.23, p<0.05). In the hippocampus, the number of fields in the choice arms versus the forced arms was not significantly different (two-tailed two sample t-test t(364) = 2.34, p>0.05), while, in the LS, there were significantly more fields in the choice arms compared to the forced arms (two-tailed two sample t-test t(334)=-3.8, p<0.001).

We wondered if place field location depending on direction of travel; for instance, if it was more likely to see a place field by a reward site after the site had been visited. To determine this, we split fields by direction, based on whether the animal was traveling to or from reward (if a field existed in both directions, we analyzed its parameters in both directions). This resulted in a total of 209 hippocampal place fields (115 toward reward and 94 away from reward, with, out of the total, 133 being unidirectional and 76 being bi directional) and 248 LS place fields (138 toward reward, 110 away from reward, with, out of the total, 177 unidirectional and 71 bidirectional. There was no significant difference of numbers of uni- or bi- directional HPC or LS cells, two-tailed two sample t-test, t(461) =1.377, p>0.05), and no difference in the distribution of HPC or LS place cell locations depending on if a cell was uni- or bi- directional both (KS test, p>0.05). There was no significant different between field sizes based on direction within the LS, within the HPC, or across the LS and HPC (all two-tailed two sample t-test p>0.05).

In the LS, the difference seen in the number of fields on the forced side versus choice side was most stark when the animal was traveling away from reward (two-tailed two sample t-test t(218)=3.8 p<0.001), although there was a clear trend while the animal traveled toward reward (two-tailed two sample t-test t(254)=1.82 p<0.1) (*Figure 6G*). In the hippocampus, there were never significantly more fields on the choice side (*Figure 6F*). While LS place fields were more concentrated around reward sites, average place field firing rate did not scale with distance to reward in either the LS or the hippocampus (*Figure 6—figure supplement 2*).

We also computed the probability of finding a place field as a function of distance from reward (*Figure 6H–I*). In the hippocampus, there was an increase in the probability of a spatial firing field in the last 60 cm of reward approach (*Figure 6H*). However, the largest peeks in HPC place field probability were around the forced and choice points of the maze, approximately 200–220 cm and 80–60 cm away from reward, respectively. In the LS, the entire forced arms were highly overrepresented, and the probability of a place field also increased upon reward approach.

In order to determine if HPC place cells were preferentially innervating LS cells, we computed spike train cross correlations between HPC and LS place cells with place fields in similar locations (centers less than or equal to 20 cm apart) (*Wilson and McNaughton, 1994*). (To adjust for spiking variance, we also computed the cross correlation with a shuffled LS spike train, and subtracted the mean of this cross correlation from the computed pairwise cross correlation.) We found that place cell pairs with fields in the choice side of the maze had a significantly higher mean average correlation (over +−100 ms) than place cell pairs with fields in the middle or forced arm (*Figure 7A*) (forced versus choice: two-tailed two sample t-test t(67)=-2.2, p<0.05, middle versus choice: two-tailed two sample t-test t(54)=-2.3, p<0.05). For pairs in the choice arm, the maximum value of the cross correlation happened at a mean lag of 20 ms with the HPC leading (*Figure 7B*, for all cross correlations, see *Figure 7—figure supplement 1*), within approximately the same time course seen for sharp wave ripple propagation from the HPC to the LS (*Wirtshafter and Wilson, 2019*). It does not appear that the higher cross correlations for cells on the choice side of the track were due to higher firing rates of cells proximal to reward, as there was no significant difference between the mean and maximum firing rates of LS place cells in all three locations (see *Figure 7—figure supplement 2*). The

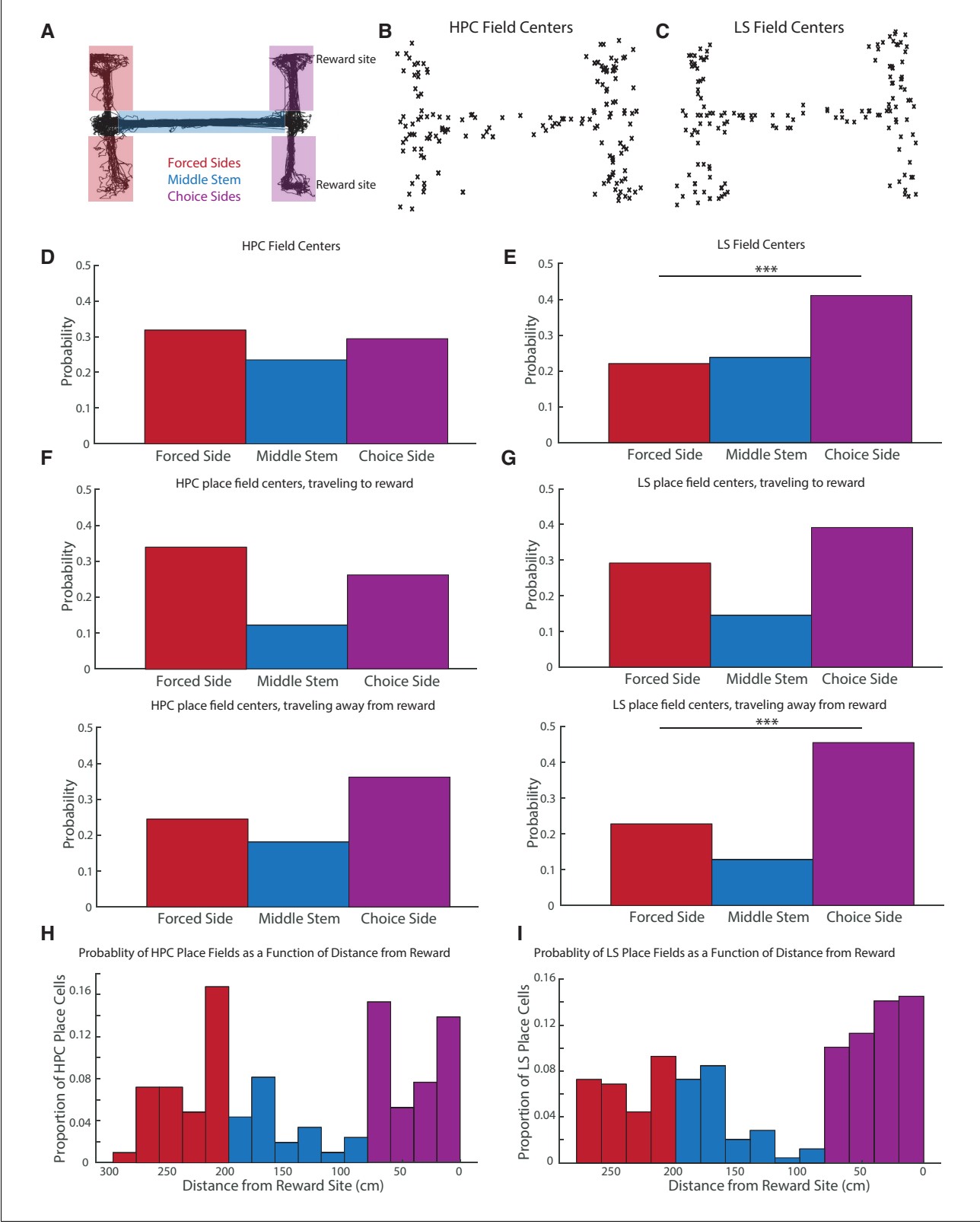

**Figure 6.** CA1 and LS fields have different location distributions, with LS place fields more biased toward reward locations. The distribution of place fields is significantly different between the hippocampus and the lateral septum (Pearson's chi2 test, $X^2$=12.7, p<0.05). (**A**) Schematic showing where the maze is split to identify place field location. (**B**) Scatter plot of HPC field centers (**C**) Scatter plot of LS field centers D. Distribution of HPC field centers. In CA1, place fields are no more likely to be on the choice side of the maze than the forced side (two-tailed two sample t-test t(364) = -2.34, p>0.05) (**E**)

*Figure 6 continued on next page*

Figure 6 continued

Distribution of LS field centers. There were significantly more fields in the choice arms compared to the forced arms (two-tailed two sample t-test t(334) =-3.8, p<0.001). The LS also has about 1.4 times, proportionally, more place fields in the choice side than the HPC does (41.1% of fields in the LS, versus 29.2% of fields in the HPC, two-tailed two sample t-test t(320) = -2.23, p<0.05). (F) Distribution of HPC place field centers by direction of travel. Top: traveling to reward. Bottom: traveling away from reward. (G) Distribution of HPC place field centers by direction of travel. Top: traveling to reward. Bottom: traveling away from reward. The LS had significantly more place fields on the choice side than the forced side in both travel directions, with the difference highly pronounced travelling away from reward (travelling toward reward two-tailed two sample t-test t(254)=1.82 p<0.10, travelling away from reward two-tailed two sample t-test t(218)=3.8 p<0.001). The distribution of place fields in the choice side was also significantly different based on the animal's travel (one-tailed two sample t-test t(102)=-2.15, p<0.05). (H) The probability of finding a hippocampal place field as a function of distance from rewarded locations. As above, red represents the forced side of the maze, blue the middle arm, and purple the choice side. Note that the divisions between the three segments of the maze are not exactly represented in the histogram due to binning. (I) Same as H but for the LS.

The online version of this article includes the following figure supplement(s) for figure 6:

**Figure supplement 1.** Subdivided place field locations in HPC and LS.
**Figure supplement 2.** Mean place field firing rate is not correlated with distance to a reward site.

higher cross correlations were also not the result of greater proximity of field centers between LS-HPC place cell pairs in the choice arm (see *Figure 7—figure supplement 3*).

Finally, we sought to determine if correlations between HPC and LS spiking became stronger throughout a single day's training (*Figure 7C*). Comparing pairs of cells within the first third, middle third, and last third of training, there were no significant differences within group (e.g. cross correlation of reward proximate cells in the first third versus the last third of training) (all paired double sided t tests, p>0.05). However, across all three periods of training, cross correlations for reward proximate cells were consistently higher than for more reward-distal cells. It therefore appears that activity correlations between HPC and LS reward-proximate cells are either developed early in training before the task is learned and not lost between training sessions, or they are not developed within training sessions.

## Discussion

In this study, we have completed a comprehensive characterization and analysis of LS place cells and their properties using parameters identical to those used to characterize HPC place cells. We found that, while the percentage of place cells is lower in the LS than in CA1, the observed LS place cells are similar to those in the CA1 in terms of field numbers and field size. We additionally determined that LS do not converge on a place field center with as much precision as CA1 cells. Finally, we found that LS place fields are distributed more toward reward locations than place fields in CA1, and that this distribution is even more biased toward rewarded locations if the animal is travelling away from the reward site. We suggest that these characteristics are the result of selective LS innervation by HPC place cells, which then allows reward related information to be sent to areas downstream of the LS.

Previous reports have varied wildly on the prevalence of place cells in the LS, with estimates ranging from none (*Tingley and Buzsáki, 2018*), to a third (*Leutgeb and Mizumori, 2002*; *Zhou et al., 1999*) to almost half or more of all cells (*Takamura et al., 2006*). We found that 26.5% of LS cells could be characterized as place fields based on both a bits per spike cutoff and spatial firing characteristics. The difference in estimates can be accounted for by several variables: first, using a stringent bits/spike and field size criteria eliminates many cells that may fit one criteria or the other, or that may have very small fields. Additionally, as we found that LS place fields tended to be clustered around reward locations, the presence, absence, and location of reward could have affected the total number of place fields. Finally, the caudodorsal LS receives the densest innervation from the hippocampus (*Swanson, 1977*), so more ventral areas of the LS may have fewer place fields, a result that has been previously observed (although place fields have been found throughout the entirety of the LS) (*Takamura et al., 2006*).

LS place field sizes are comparable to HPC field sizes, and both HPC and LS cells increase place field spiking with experience. However, LS place cells are slightly but significantly less accurate than HPC place fields (*Figure 4E–H*). This may be the result of LS cells incorporating other parameters beyond current spatial location. We have previously shown that LS cells' firing rate may be affected

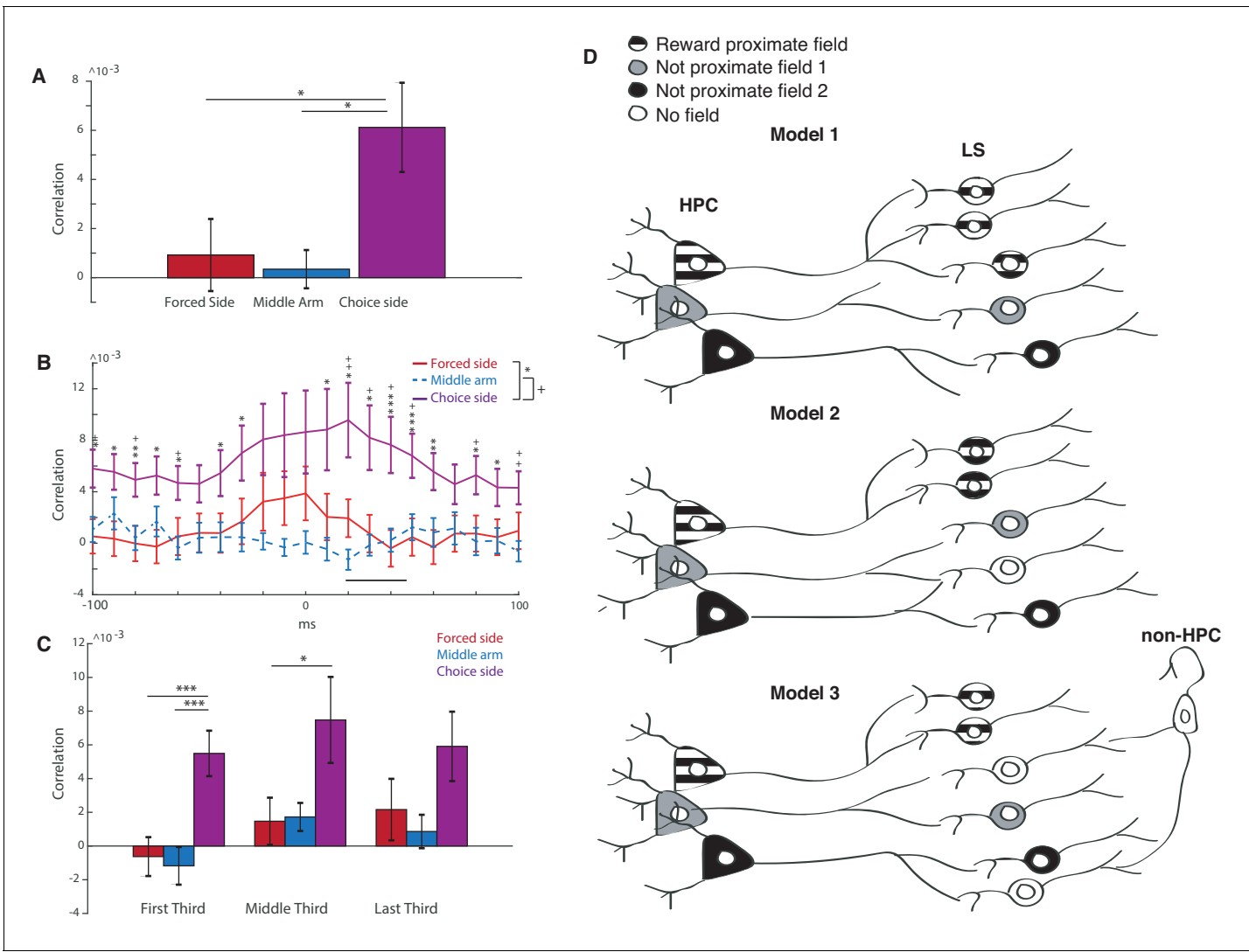

**Figure 7.** Reward proximate LS cells are more synced with hippocampal activity than not proximate cells. (**A**) Averaged cross correlation across a lag of −100 to 100 ms between coupled pairs of HPC and LS cells, based on place field location. Error bars show standard error. Forced side versus choice side: two-tailed two sample t-test t(67)=-2.2, p<0.05, middle arm versus choice side: two-tailed two sample t-test t(54)=-2.3, p<0.05). (**B**) Average of cross correlation traces at all lags between coupled pairs of HPC and LS cells, based on place field location. Error bars show standard error. Mean correlation peak for forced arm correlations was at 20 ms. Results from unpaired two tailed t-tests between forced and choice arms are indicated with stars: * indicates a p value of < 0.05 on an unpaired two tailed t-test, ** indicates a p value of < 0.01, and *** indicates a p value of < 0.005. Analogous results are shown between choice arm and middle arm using +. (**C**) Differences in cross correlations for HPC-LS pairs during the first, middle, and last third of a single trial. Error bars show standard error. * indicates a p value of < 0.05 on an unpaired two tailed t-test, *** indicates a p value of < 0.005. (**D**) Three models of LS innervation by HPC that may explain the overrepresentation of LS place fields by rewarded locations. Top: model 1. HPC cells with reward-proximate place fields selectively innervate more LS place cells. Middle: model 2. HPC cells with reward proximate place fields, and cells with other fields innervate the same number of LS cells. Hippocampal cells with non-proximate fields innervate overlapping cells, causing interference and resulting in fewer place fields in the LS that are not reward proximate. Bottom: model 3. HPC cells with reward proximate place fields, and cells with other fields innervate the same number of LS cells. Hippocampal cells with non-proximate fields innervate cells that are also innervated by other inputs, causing interference and resulting in fewer place fields in the LS that are not reward proximate.

The online version of this article includes the following figure supplement(s) for figure 7:

**Figure supplement 1.** Cross correlation between all coupled pairs of HPC and LS cells, based on place field location.

**Figure supplement 2.** Mean and maximum within-field firing rates for LS place cells based on location.

**Figure supplement 3.** Differences in cross correlations are not due to differences HPC-LS place field pair proximity in the different maze arms.

by an animal's speed or acceleration (*Wirtshafter and Wilson, 2019*), which may decrease the precision of place field firing. The incorporation of proximity to reward location, discussed below, may also serve to modulate LS place cell firing beyond the animal's current location.

We observed that the distribution of place fields in the lateral septum was more biased toward the rewarded locations of the maze than the distribution of place fields in CA1 (*Figure 6*), and that, unlike HPC place fields, LS place fields tended to skew toward reward direction regardless of the direction of travel, particularly when close to rewarded locations (*Figure 5*). Previous work has also established that the LS can represent both reward delivery and reward locations (*Nerad and McNaughton, 2006*; *Wirtshafter and Wilson, 2019*). There are many possible ways this result could be accounted for at the circuit level: First, reward information may be introduced into the lateral septum from an area outside of the hippocampus. The LS has connections, often reciprocal, with many brain areas that respond to reinforcing stimuli, including the VTA (*Gomperts et al., 2015*; *Jiang et al., 2018*; *Luo et al., 2011*; *Roesch et al., 2007*), hypothalamus, which is involved with the LS in a circuit that regulates feeding (*Sweeney and Yang, 2016*), ventral hippocampus (*Zhou et al., 2019*), and amygdala (*Krettek and Price, 1978*; *Sheehan et al., 2004*). It is possible that information about reward location is being sent from, for example, the VTA to the LS, rather than vice versa. However, we think it unlikely that this circuit accounts for the all, or the majority, of reward proximate place firing in the LS for several reasons: First, reward related cells in the VTA are specifically reactivated during hippocampal replay and sharp wave ripples at a time course consistent with the signal first travelling through the LS from the hippocampus (*Gomperts et al., 2015*; *Wirtshafter and Wilson, 2019*). Additionally, cells in the LS that change firing during a reward period tend to show signatures of hippocampal interactions, such as theta coherence and increased firing during hippocampal sharp-wave ripples (*Wirtshafter and Wilson, 2019*). Finally, we found that LS cells with reward proximate firing are more highly cross correlated with hippocampal firing than cells without reward proximate place fields (*Figure 7*). This suggests that a predominant input to these LS cells that are biased to reward locations is from the hippocampus.

In the present task, the probability of finding a HPC place field increases as the animal approaches final goal location (*Figure 6H*). However, the representation of the goal is decreased relative to locations proximate to the forced and choice points of the maze. Past work has found that the HPC uniquely over-represents salient or goal locations on tasks demanding increased spatial memory (*Dupret et al., 2010*). It is possible, therefore, since the important spatial memory components of this task occurred at the forced and choice points, that these locations came to be even more over-represented in the HPC that the goal locations.

Reward-related spatial information may be transmitted from the HPC to the LS in multiple ways. First, there may be a fixed population of HPC cells that consistently represent reward, which selectively innervate the LS. There is direct evidence for such a population (*Gauthier and Tank, 2018*), but it is unknown if it comprises a large portion of LS afferents. Alternatively, different HPC place cells may represent reward in different spatial environments, but the LS cells receiving place cell input are differentially modulated by external inputs depending on the specific task.

So how are reward proximate place cells more common in the LS than in the hippocampus? First, it is possible that hippocampal place cells that have fields proximate to reward locations selectively, either anatomically or functionally, innervate LS cells (*Figure 7D*, Model 1). Additionally, it is possible that non-reward related place cell inputs from the hippocampus are diluted in the LS if they converge with non-reward related input. This input may be from other place cells without reward proximate fields (*Figure 7D*, Model 2), or from other areas, such as speed or acceleration related information from the brainstem (*Figure 7D*, model 3; *Wirtshafter and Wilson, 2019*). In this model, reward proximate place cells would innervate a separate group of LS cells which would account for the over-representation of reward proximate place fields in the LS (*Figure 7D*, Model 2). Our finding that LS and HPC place cells with fields proximate to reward have increased cross correlations compared to cells with more distal place fields is consistent with any of these models, as the higher cross correlation may be due to a selective innervation of the LS by reward-proximate HPC cells, or the 'dilution' of non-reward related place cells by competing inputs (*Figure 7D*, Model 3). However, our previous work (*Wirtshafter and Wilson, 2019*) has shown that LS cells containing place information are less likely to contain speed or acceleration information than cells without place fields, suggesting that speed or acceleration information may be competing with spatial information sent from the HPC, a result most easily accommodated by model 3.

We also found differences in place cell activity based on whether the animal was moving toward or away from the reward site. We first found that the reward site itself was overrepresented in LS place fields right after the reward site had been visited (*Figure 6G*). We also found that during reward approach, within a place field, LS place cells, on average ramped up firing (*Figure 5E*). It is possible that both this overrepresentation and difference in skew may serve memory or planning functions during navigation. We have previously found that the HPC-LS coherence increased during working memory tasks (*Wirtshafter and Wilson, 2019*), and increased place field activity at a site of significance could be evidence of hippocampal and LS coordination during encoding of significant task locations. There is increased VTA reactivation of reward related cells during ripples and VTA cells also ramp up firing during reward approach (*Gomperts et al., 2015*), showing that the significance of reward related locations has been established downstream of the LS. That information may be communicated to the VTA from the LS by the over-representation of just-visited reward related locations in place cell firing, as well as by increased within-field firing upon reward approach.

The distribution of field numbers was not significantly different between HPC and CA1 cells (two sample Kolmogorov-Smirnov (KS) test, p>0.5).

Top panel is traveling to reward, bottom panel is traveling away. Because of the non-linearity of the track, different place fields were visited different numbers of times, and therefore each field contributes a different number of bins to these histograms.

## Materials and methods

### Subject details

All procedures were performed within MIT Committee on Animal Care and NIH guidelines. Six male Long-Evans rats (275 g to 325 g) were sourced from Charles River and implanted with tetrode arrays and run on a double-sided T maze (see *Figure 1*). Different data from the same subjects were previously published in *Wirtshafter and Wilson, 2019*. Animals were individually housed in an animal facility with a 12 hr light dark cycle.

### Method details

#### Tetrode implementation and electrophysiology

Rats were implanted under isoflourine anesthesia (induction 4%, maintenance 1–2%) with two multitetrode arrays, each containing 16 independently moveable tetrodes (see *Jones and Wilson, 2005b*). One tetrode array was directed toward the dorsal CA1 hippocampus (stereotaxic coordinates Bregma −3.7, midline −3.2), while the other was directed toward the caudo-dorsal lateral septum (stereotaxic coordinates Bregma +.05, midline −0.5). Animals were grounded with a skull screw posterior to Lambda. Over several days tetrodes were individually lowered to their goal location. A CA1 reference tetrode was placed in the corpus collosum white matter tract above CA1. A lateral septum reference tetrode was placed in white matter above the LS, or at a quiet site in the lateral septum.

Electrical signals were passed through two 16 channel headstage preamplifiers to a custom patchbox which was then used to select a reference channel. The signal was then fed to Neurolynx amplifiers. Extracellular action potentials were acquired at 31 kHz, 0.3–6 kHz filtering. Data were collected using custom lab software ARTE (*Hale and Wirtshaftee, 2019*, June 30). The animal's position during trial runs was collected at 30 Hz via overhead cameras. Position was collected and later extracted using OAT (*Newman et al., 2017*, December 10).

After data collection, CA1 and LS cells were manually isolated using a custom software package (Xclust) using spike amplitude on each of the four channels. Septal and hippocampal cells with large amounts of drift and/or unstable waveforms were excluded in the analysis. After completion of the study, animals were lesioned with 15μA of current for 10 s to mark tetrode location. Animals were then perfused at least one week post lesioning, and tetrode locations were then verified with histology.

#### Behavioral training

During training, animals were food deprived to 85% body weight. Implanted animals were trained for 2–4 weeks to run a spatial choice task (*Gomperts et al., 2015*; *Jones and Wilson, 2005b*) on an

end-to-end T maze (*Figure 1*). Animals were free running on this task and were only handled to be placed on the maze and removed from the maze. The maze had two phases: in the forced choice phase, animals were randomly forced to either side of the T into the forced arms. (Animals were not directed to one side more than three consecutive times.) In the free choice phase, animals had to choose, at the opposite end of the maze, the same side that they were forced to in the free choice arm. If animals made the correct choice, animals were rewarded with 0.2 mL of 20% sucrose 10% chocolate milk powder dispensed remotely from a syringe pump. After a trial, animals self-initiated a new trial by returning to the forced arm of a maze. Tail tip in an arm was used as the criteria for arm entrance. Animals were trained to a criterion of 75% correct choices. There was wide variability among all the animals for the length of time it took to learn the task, as well as their ability to maintain their performance at criterion. Animals were each run for 30 min a day.

## Data and code availability

All analysis code was custom written and anaylysis was performed using Matlab (MathWorks, Natick, Massachusetts). Code is public on https://github.com/hsw28/data_analysis/ (*Wirtshafter, 2020*; copy archived at https://github.com/elifesciences-publications/data_analysis).

## Statistical analysis

### Position and velocity sampling

Position was sampled by overhead cameras at about 30 Hz. Due to occlusion, sampling rates of position were often at approximately 15 Hz. Speed was determined by taking the hypotenuse of the coordinates of the points immediately before and after the time point of interest. Speed was then smoothed using a Gaussian kernel of 1 s standard deviation and was then converted from pixels/s to cm/s.

### Bits/spike and bits/second

All animal occupancies and spikes were found when the animal was travelling at or over 12 cm/second. Firing per occupancy found in 1 cm$^2$ increments, smoothed with 10 cm standard deviation Gaussian kernel.

A bits per spike measurement was calculated as follows:

$$i = spatial\ bin\ number$$

$$P_i = occupancy\ probability\ for\ bin\ i$$

$$R_i = mean\ firing\ rate\ at\ bin\ i$$

$$R = overall\ mean\ firing\ rate$$

$$\text{bits per spike} = \sum_i \left( P_i \left( \frac{R_i}{R} \right) \log_2 \left( \frac{R_i}{R} \right) \right)$$

A bits per second measurement was calculated as follows:

$$\text{bits per second} = \sum_i \left( P_i \left( R_i \right) \log_2 \left( \frac{R_i}{R} \right) \right)$$

Cells were discarded if mean rate was below 0.05 hz during the time period above velocity threshold. Mutual information was found as in *Ego-Stengel and Wilson, 2007*.

### Place field analysis

Velocity was measured as determined above, and threshold was set at 12 cm/second. All animal occupancies and spikes were found when the animal was travelling at or over 12 cm/second. Firing per occupancy was found in 2 cm increments, smoothed with 10 cm standard deviation Gaussian kernel.

A place field was identified as a region of connected bins (connected on at least one of four sides, not diagonally) where firing rate was equal to or less than one standard deviation greater than the mean firing rate. In order to be classified as a place field, firing in at least one location in this region must be greater than two standard deviations greater than the mean firing rate. Place field area must be at least 15 cm long. We analyzed place field selection in our HPC data using these criteria and criteria previously established in *Rich et al., 2014*. We found that out criteria identified place fields in the hippocampus consistent with the previously established criteria (*Rich et al., 2014*).

Cells with a max rate of less than 0.05 hz were discarded from analysis.

Place field centers were determined by the coordinates within the place field that had the highest spiking rate.

Directionality was determined by computing place fields in both directions. If a unit had fields in both directions with centers separated less than 20 cm, the field was considered bidirectional. For bidirectional place cells, skew was computed in both directions.

In lap by lap analysis, place fields were included if the field was passed through at least 15 times, and laps were included if at least 20 place fields had that many laps.

Skew was determined as the ratio of the third moment of the place field firing rate spatial distribution found in the direction of travel, divided by the cube of the standard deviation of this distribution (*Spiegel, 1961*). Because skew was determined lengthwise along the track, firing rates along the width of the place field were averaged.

FRAI (*Mehta et al., 2000*) was computed as
$F_1$ = mean firing rate for first half of spikes in field
$F_2$ = mean firing rate for second half of spikes in field
$$FRAI = F_1 \, F_2 \, / \, (F_1 + F_2)$$

## Cross correlation analysis

Place cells from HPC were matched with place cells from the LS recorded during the same session based on field firing center (if a cell had multiple fields it was matched multiple times). In order to be matched, firing field centers must have been within 20 cm of each other (*Wilson and McNaughton, 1994*). If there were multiple matches, the match with the most similar average firing rate was chosen. If no matches could be found, the cell was excluded from analysis.

Cross correlations of HPC spike train x LS spike train were determined between all track spiking for matched cell pairs. Spike trains for cells were found in 10 ms bins and cross correlations were taken over $+-100$ ms. For control, an additional cross correlation was determined with HPC spike train x shuffled LS spike train. This control was subtracted from the previously determined cross correlation to get the final cross correlation for analysis.

## Acknowledgements

HSW was supported by the Department of Defense (DoD) through the National Defense Science and Engineering Graduate Fellowship (NDSEG) Program. We thank Israel Donato Ridgley and Molly Quan for support with analysis and experiments.

## Additional information

### Funding

| Funder | Grant reference number | Author |
| --- | --- | --- |
| U.S. Department of Defense | NDSEG Fellowship | Hannah S Wirtshafter |

The funders had no role in study design, data collection and interpretation, or the decision to submit the work for publication.

### Author contributions

Hannah S Wirtshafter, Conceptualization, Data curation, Formal analysis, Funding acquisition, Validation, Investigation, Methodology, Writing - original draft, Writing - review and editing; Matthew A Wilson, Supervision, Funding acquisition, Project administration, Writing - review and editing

### Author ORCIDs

Hannah S Wirtshafter (iD) https://orcid.org/0000-0003-4684-7074

### Ethics

Animal experimentation: All procedures were performed within MIT Committee on Animal Care and NIH guidelines under Wilson protocol 0417-037-20. All surgeries were done under isoflourine anesthesia (induction 4%, maintenance 1-2%) and every effort was made to minimize suffering.

### Decision letter and Author response

Decision letter https://doi.org/10.7554/eLife.55252.sa1
Author response https://doi.org/10.7554/eLife.55252.sa2

## Additional files

### Supplementary files

• Transparent reporting form

### Data availability

Data has been deposited to Collaborative Research in Computational Neuroscience (CRNRS) under the accession code hc-29 (https://doi.org/10.6080/K0NG4NV8). Users must first create a free account (https://crcns.org/register) before they can download the datasets from the site. All analysis code is available at https://github.com/hsw28/data_analysis (copy archived at https://github.com/elifesciences-publications/data_analysis).

The following dataset was generated:

| Author(s) | Year | Dataset title | Dataset URL | Database and Identifier |
|---|---|---|---|---|
| Wirtshafter HS, Wilson MA | 2020 | Tetrode recordings of hippocampus CA1 and dorsal lateral septum in rat. | https://doi.org/10.6080/K0NG4NV8 | Collaborative Research in Computational Neuroscience, 10.6080/K0NG4NV8 |

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
