## [Decision Letter]

**Acceptance summary:**

There are still so few studies about the lateral septum, disproportional to its potential significance. This is solid work with a novel finding that increases our understanding of mechanisms underlying motivated behaviors and neural representations of space.

**Decision letter after peer review:**

Thank you for submitting your article "Differences in reward biased spatial representations in the lateral septum and hippocampus" for consideration by *eLife*. Your article has been reviewed by Laura Colgin as the Senior Editor and Reviewing Editor and three reviewers. The following individual involved in review of your submission has agreed to reveal their identity: Alexey Ponomarenko (Reviewer #1).

The reviewers have discussed the reviews with one another and the Senior Editor has drafted this decision to help you prepare a revised submission.

Wirtshafter and Wilson investigated spatial firing of lateral septal (LS) and CA1 neurons using parallel recordings from the two regions during a rewarded navigation task. They show that a large number of LS cells display place-selective firing, a finding the authors have previously reported. Here, they perform more detailed analysis of place field characteristics of LS neurons, showing that LS place fields are similar in many ways to HPC fields and that LS place fields are more likely to occur near reward/choice locations. They found that one-dimensional place fields of LS neurons are in many respects similar to CA1 place fields, yet they display different skewness in relation to reward location. Further, firing of LS and CA1 cells is more correlated for those cells with place fields close to the reward location. Because neural representations in LS are poorly understood, studies of circuits including LS can provide new insights into functions of hippocampus and other interconnected regions. The present high-quality data contribute to the understanding of information processing in LS and substantially extend earlier reports (Takamura et al., 2016) showing the influence of reward location on spatial firing of LS neurons. The LS has been given somewhat short shrift by the broader hippocampal community, and although the work is somewhat exploratory, this was viewed as appropriate for the relatively early days of investigating the LS.

However, the study was viewed as having some shortcomings in its quantifications that leave open important questions. The paper needs to show a more transparent presentation of the results. The results should also be better situated in the existing literature.

Essential revisions:

1) There was a considerable variability of firing in individual runs, and LS place fields maxima did not converge to a single location. Reviewers had questions about the variability in LS firing patterns:

a) Was this variability in the different arms of the maze, and other features of spatial representations in general, related to behavioral performance or behavioral variations in the task?

b) Related to the above point, an underlying assumption is that LS neurons are selectively encoding information about reward proximity or recent reward receipt (supported by the data in Figure 6). The authors observe a higher probability of LS place fields on the Choice Side when the rat is leaving vs. approaching a reward well. An open question which is important to interpret these results is whether LS neurons care only about location relative to *possible* reward, or rather care about location relative to *actual* reward. Given that rat performance is ~75% correct according to the methods, do the authors observe an effect of success or failure on the task? Do LS neurons differentiate between runs toward or away from the same location when that location is actually rewarded vs. when it was not? On a similar note, given the increase in LS place fields upon leaving the reward site, do the authors observe larger LS place fields after correct vs. incorrect trials? Alternatively do the authors observe LS place fields which do not distinguish between the two arms, but instead only represent correct vs. incorrect trials?

c) There was confusion about the results described in Figure 4E-F. The fact that the curve for hippocampal neurons eventually reaches zero indicates that spatial representation stabilizes over time. However, while the curve is downward for LS cells (indicating some level of stabilization), the reason why it never reaches zero was unclear. Do these data simply mean that each lap is highly variable for LS cells even after a small amount of initial stabilization? If so, and if the final distance of lap place field center to average place field center is 20 cm, this indicates that the 'place field' on each lap moves up to 40 cm (+/- 20 cm from average center). Considering that the track center is ~120 cm long and each arm is ~70 cm long, movement of 40 cm from lap-to-lap seems quite substantial. Is it possible that LS place fields shift slowly across time, making it near-impossible for any given lap to align with the average across all laps? Given the bias for LS place fields to be located near the reward well, are near-reward place fields more or less likely to stabilize than reward-distant fields?

d) LS cells are quite heterogeneous in their firing rate and the regularity of discharge. Was there any systematic relationship between spatial firing properties and firing rate, coefficient of variation and the recording location? This information may also help explain apparent discordances with other studies of spatial firing in LS (Tingley and Buzsaki, 2018).

2) Regarding the data in Figure 2, the distribution of HPC unit bits/spike appears to be comprised of at least two (possibly three) clusters: one with very low bits/spike (near zero), one with moderate bits/spike (centered around 1.25), and possibly a third with very high bits/spike (centered around 2.9). This is what would be expected in a population of neurons in which most, but not all, encode spatial information in any given environment. Appropriately, the division between cells that encode spatial information and those that do not (bits/spike = 0.8) does seem to correctly divide those populations. However, for the LS neurons, the distribution appears to be composed of a single population, and the separation that the authors make between cells that encode spatial information and those that do not seems rather arbitrary for LS cells. In other words, if you only looked at the distribution of bits/spike for LS cells, where would you draw the red dashed line? Reviewers raised a concern that the distribution of bits/spike for LS cells is not reflective of a true spatial encoding, but rather, what one might expect from random firing patterns. If the authors shuffle cell IDs for each spike, do they still observe a similar distribution of bits/spike across the shuffled units? If the authors create 452 artificial units using Poisson firing, do they observe a similar distribution of bits/spike for the artificial units? In other words, is the place representation of some neurons really different than one would expect from chance, given that the authors are sampling 452 neurons and at least some of those neurons are likely to fire in spatially restricted locations by chance?

3) Could increased cross-correlations computed for a broad range of lags of 100 ms in the choice arm be due to higher firing rates proximal to rewards? Information about firing rates (peak rates, average rates in field) in different arms is difficult to find in the manuscript.

4) Given the importance of LS place fields to this study, reviewers would like to see more than four examples. Reviewers suggested a Supplemental Figure that presents a large number of LS place fields, selected in an unbiased way.

5) Reviewers would also like to see examples of spike cross-correlations supporting the data in Figure 7.

6) The authors note in the introduction that the hippocampal encoding of goal locations has been characterized, but they do not cite a highly-relevant study: Dupret et al., 2010. These experiments showed that the hippocampal over-representation of goal is not universal, but instead depends on the cognitive demands of the task. It is important to know, therefore, whether goal locations are over-represented in the present study. One straightforward way to address this would be to re-analyze the data from panels 6E-F. If the probability of a field were *per unit distance*, results from the different track segments would be directly comparable. This would reveal whether hippocampal place cells themselves are clustered near reward in this task, and to what degree LS neurons might amplify that clustering.

7) The plot in Figure 7B shows an intriguing correlation. This quantification only shows the effect is present when averaging across all neurons, however. Does every neuron show such a correlation, or just a subpopulation? This information could help to corroborate or reject the authors' models. Related to the previous point, it would also provide an estimate of what fraction of hippocampal neurons might be specialized for encoding reward.

8) The recording location in the caudodorsal LS is very close to the septohippocampal nucleus (SHN) based on Paxinos and Watson's stereotaxic atlas, and indeed, at least one of the recording sites in Figure 1A appears to be in the SHN. I am unaware of any literature quantifying place representation in the SHN, but given the controversy regarding whether LS neurons have place fields, it is important to determine whether the place-selective units in the current study are more likely to be LS or SHN neurons. Do the authors observe a correlation between medial/lateral (or other stereotaxic orientation) recording location and likelihood of observing place-specific firing? Perhaps in a supplemental analysis, the authors could replicate several of their core findings using only units recorded on the lateral-most tetrodes, which would be the least likely to be in the SHN.

9) In Figure 5, the authors quantify skewness and compare runs to reward vs. runs away from reward, reporting a significant difference for hippocampal fields, but not for LS fields. To facilitate interpretation of these results, reviewers would like to know if the skewness of HPC or LS neurons during runs to reward were different from zero (or different from a shuffle distribution). The same point was raised for runs away from reward.

10) Do LS cells have direction-specific place fields, and how does direction-selective firing (in HPC and LS neurons) impact skewness? Is the skewness driven entirely by uni-directional fields, or is it observed for both uni- and bi-directional fields?

11) The datapoints in Figure 5 and Figure 5—figure supplement 2 don't seem to line up, and there was confusion about these figures. Is it correct that each dot represents a cell with its average place field peak at that location (relative to reward) and with that skewness? If that is correct, shouldn't the data in Figure 5F-G be a subset of the data in Figure 5—figure supplement 2? However, those points aren't the same. The difference should be made clear in the text or in the figure legend.

[Editors’ note: further revisions were suggested prior to acceptance, as described below.]

Thank you for resubmitting your article "Differences in reward biased spatial representations in the lateral septum and hippocampus" for consideration by *eLife*. Your article has been reviewed by Laura Colgin as the Senior Editor and Reviewing Editor, and three reviewers. The following individuals involved in review of your submission have agreed to reveal their identity: Alexey Ponomarenko (Reviewer #1).

The reviewers have discussed the reviews with one another and the Reviewing Editor has drafted this decision to help you prepare a revised submission.

Summary:

Reviewers agree that authors have satisfactorily addressed most concerns. Reviewers are grateful that the authors identified the coding error that led to many of the reviewer questions and also appreciate the additional work the authors have performed on analyses and reworking the text of the manuscript. Reviewers agree that this is a considerably stronger manuscript after the revisions. There are only a few comments remaining:

Essential revisions:

1) It is still possible that the higher cross-correlations over many tens of milliseconds between HPC and LS cells with place fields in the choice side (Figure 7) can be secondary to a higher number of place fields in both regions in this part of the maze (Figure 6). This might be addressed, for instance, by comparing cross-correlations for forced, middle and choice sides for subsampled spike trains selected to ensure a matched degree of overlap of place fields in HPC and LS. However, the paper would still be interesting and worthy of publication even if it turns out that cross-correlations are due to concentration of Hip and LS place fields on the choice side close to rewards.

2) Figure 5 seems to be consistent with the lack of influence of hippocampal inputs on the skewness of LS place fields. The latter are similarly skewed at different locations. However, HPC place fields change their directional skewness depending on the proximity of the reward. It would be helpful to integrate this finding in the models proposed.

---

## [Author Response]

Essential revisions:1) There was a considerable variability of firing in individual runs, and LS place fields maxima did not converge to a single location. Reviewers had questions about the variability in LS firing patterns:

The high amounts of field center variability/maxima originally seen in Figure 4E-F were reevaluated and we discovered a minor computational error involving cells with multiple fields. That mistake has been fixed and all results in Figure 4E-F have been redone, with the addition of Figure 4G-H. We discuss this more in response to question 1C below, but, in brief, the average distance from a place field center for peak LS firing has now been found to be 5.12cm, with a standard deviation of +-7.1cm (as opposed to the >20cm of distance from field center previous reported). This correction does not influence our overall conclusions regarding LS place fields.

a) Was this variability in the different arms of the maze, and other features of spatial representations in general, related to behavioral performance or behavioral variations in the task?

As explained above, the variability is much smaller than originally reported, and does not appear to differ across arms of the maze. Because LS cells are known to incorporate speed and acceleration information in spike rate, we attempted to take into account this behavioral variation by computing a linear regression of spike rate against speed and acceleration for all LS cells with a bits/spike over 0.8. However, very few cells had large r2 values (Figure 3—figure supplement 1). This is consistent with our previously published work which shows that cells that code for spatial location tend not to also code for acceleration or speed. We have made this point more clear in the text.

b) Related to the above point, an underlying assumption is that LS neurons are selectively encoding information about reward proximity or recent reward receipt (supported by the data in Figure 6). The authors observe a higher probability of LS place fields on the Choice Side when the rat is leaving vs. approaching a reward well. An open question which is important to interpret these results is whether LS neurons care only about location relative to *possible* reward, or rather care about location relative to *actual* reward. Given that rat performance is ~75% correct according to the methods, do the authors observe an effect of success or failure on the task? Do LS neurons differentiate between runs toward or away from the same location when that location is actually rewarded vs. when it was not? On a similar note, given the increase in LS place fields upon leaving the reward site, do the authors observe larger LS place fields after correct vs. incorrect trials? Alternatively do the authors observe LS place fields which do not distinguish between the two arms, but instead only represent correct vs. incorrect trials?

We attempted to do this analysis prior to the original submission, but unfortunately the bounds of the task make it rather hard. With between 20 and 30 trials a day, the animal is likely only to visit an individual rewarded arm between 10 and 15 times. if the animal is performing at above 75%, that often meant the animal passed through a place field two or fewer times per day on incorrect trials on their way to reward. Unfortunately, that did not provide a sufficient sample to determine whether individual cells were responding to the presence or absence of reward versus reward expectation, or the effect of reward receipt on place cell characteristics.

We did examine whether some cells appeared to represent rewarded locations preferentially. To do this, we looked at all cells that had more than one place field, and determined if the odds of having two place fields in the rewarded arms were higher than having a place field at a rewarded arm and a choice arm or the middle stem.

Out of the total number of cells with multiple place fields, 23 of these cells had exactly two place fields that occurred on one of the stems (as opposed to the choice points). Out of the corresponding 46 fields, 11 (23.9%) occurred in the forced arm, 12 (26.1%) occurred in the middle stem, and 23 (50.0%) occurred in the choice side, with 11 (23.9%) on one side of the arm and 12 (26.1) on the other. In any pair of fields, therefore, expect a cell with a place field on each end of the forced side about 12.48% of the time. Out of the 23 place field pairs, we see this combination 5 times, or 21.74% of the time. While this is suggestive, the low number of pairs makes the result not significantly different from expected so we did not report it in our results.

c) There was confusion about the results described in Figure 4E-F. The fact that the curve for hippocampal neurons eventually reaches zero indicates that spatial representation stabilizes over time. However, while the curve is downward for LS cells (indicating some level of stabilization), the reason why it never reaches zero was unclear. Do these data simply mean that each lap is highly variable for LS cells even after a small amount of initial stabilization? If so, and if the final distance of lap place field center to average place field center is 20 cm, this indicates that the 'place field' on each lap moves up to 40 cm (+/- 20 cm from average center). Considering that the track center is ~120 cm long and each arm is ~70 cm long, movement of 40 cm from lap-to-lap seems quite substantial. Is it possible that LS place fields shift slowly across time, making it near-impossible for any given lap to align with the average across all laps? Given the bias for LS place fields to be located near the reward well, are near-reward place fields more or less likely to stabilize than reward-distant fields?

As a consequence of the error reported above, we re-computed the results for place cells with multiple fields. These new results give us a much clearer picture of highly accurate place fields in the LS and HPC, with increased firing at the place field center with experience. These results do not effect our overall conclusion that the LS is less tuned to space than the HPC. We have redone Figure 4E-F and also added Figure 4G-H. We have explained them in the text as follows:

“Previous work has demonstrated that hippocampal place fields stabilize and become more tuned to position with experience (Mehta, Barnes and McNaughton, 1997). We compared the time periods over which HPC and LS place fields become stable. For each pass through a place field, we determined how far the center (determined by maximum spiking) of the place field was from the average place field center. Both the hippocampus (Figure 4E) and LS (Figure 4F) had highly accurate place fields starting with the first pass of the place field, though, on average the HPC is slightly more accurate on the first lap as well as across all laps. (On the first lap, HPC has a mean distance of 2.7cm versus 4.9cm for the LS, two-tailed two sample t-test t(319)=-3.1, p<0.005. Across all laps, mean distance of 4.70cm for the HPC versus 5.12cm for the LS, two-tailed two sample ttest t(6792)=-2.5, p<0.05). The slight but significant decrease in accuracy from the first to later laps in the HPC can be explained by a slight shift of the place field peak towards the direction of travel (Figure 4G). Fields in both the HPC and the LS significantly increase firing in their place field with experience on the track (Figure 4G-H).”

d) LS cells are quite heterogeneous in their firing rate and the regularity of discharge. Was there any systematic relationship between spatial firing properties and firing rate, coefficient of variation and the recording location? This information may also help explain apparent discordances with other studies of spatial firing in LS (Tingley and Buzsaki, 2018).

For demonstration of the rate variability, we have added Figure 3—figure supplement 2 which shows more examples of place fields and also shows the distribution of firing rates. In answering review concern #3 we also compared mean and maximum firing rate across different track segments and found no significant difference (Figure 7—figure supplement 2). There thus appeared to be no significant relationship between firing rate and the rat’s location in the maze.

As far as tetrode location could be mapped, we saw no difference in firing rate and place cell probability from more lateral to more medial locations in the LS. We recorded the vast majority of cells in the most dorsal area of the LS so, similarly, the vast majority of place fields were located there. We found that cells appeared to be sparser, as judged by number of cells picked up by a tetrode, in more ventral LS areas.

Previous studies that recorded from the entire LS (dorsal to ventral) reported place cells along the entire depth of the LS. (For example, see Takamura et al., 2006, Figure 6:)

Examples of studies finding LS place cells in areas other than the most dorsal LS include:

- Bezzi et al., 2002

- Kita et al., 1995

- Leutgeb and Mizumori, 2002

- Monaco et al., 2019

- Nishijo et al., 1997

- Takamura et al., 2006

- Zhou et al., 1999 among others (all of which are referenced in text), so we do not think tetrode location can explain the discrepancies with the unique results found in Tingley and Buzsaki, 2018.

2) Regarding the data in Figure 2, the distribution of HPC unit bits/spike appears to be comprised of at least two (possibly three) clusters: one with very low bits/spike (near zero), one with moderate bits/spike (centered around 1.25), and possibly a third with very high bits/spike (centered around 2.9). This is what would be expected in a population of neurons in which most, but not all, encode spatial information in any given environment. Appropriately, the division between cells that encode spatial information and those that do not (bits/spike = 0.8) does seem to correctly divide those populations. However, for the LS neurons, the distribution appears to be composed of a single population, and the separation that the authors make between cells that encode spatial information and those that do not seems rather arbitrary for LS cells. In other words, if you only looked at the distribution of bits/spike for LS cells, where would you draw the red dashed line? Reviewers raised a concern that the distribution of bits/spike for LS cells is not reflective of a true spatial encoding, but rather, what one might expect from random firing patterns. If the authors shuffle cell IDs for each spike, do they still observe a similar distribution of bits/spike across the shuffled units? If the authors create 452 artificial units using Poisson firing, do they observe a similar distribution of bits/spike for the artificial units? In other words, is the place representation of some neurons really different than one would expect from chance, given that the authors are sampling 452 neurons and at least some of those neurons are likely to fire in spatially restricted locations by chance?

We chose to use a 0.8bits/spike cutoff for several reasons: first, we wanted to use the same criterion to judge the spatial firing properties of LS cells as to judge the spatial firing properties of HPC cells. Secondly, previously literature examining spatial firing in nonhippocampal areas, such as in the visual cortex, used a 0.8bits/spike cutoff for spatial firing (see Ji and Wilson 2007), and we have added a mention of this to the text. Thirdly, we wished to predefine the criterion for place cell inclusion prior to data analysis to obtain the most objective results.

To address the concerns of the reviewers, we created artificial units with Poisson firing, matching the units average firing rate to actual LS units. We added this data to figure 2B, and added the following in the text:

“To ensure that the representation of space was different than would be expected from random Poisson firing, we created 454 artificial LS units using Poisson firing and the mean firing rates of the recorded LS units (Figure 2B, inset). The distribution of the bits/spike for the artificial units was highly significantly different than the distribution of bits/spike for actual units (KS test, p<10^-15^), and only 7.96% of artificial units had bits/spike measurements of greater than 0.8. The average bits/spike for the artificial units was 0.43, compared to an average value of 0.73 for actual units (two-tailed two sample ttest, t(818)=6.72, p<10^-10^).”

3) Could increased cross-correlations computed for a broad range of lags of 100 ms in the choice arm be due to higher firing rates proximal to rewards? Information about firing rates (peak rates, average rates in field) in different arms is difficult to find in the manuscript.

This seems unlikely, as the middle arm actually had the highest mean and maximum firing rate per place field and the lowest cross correlation, and the mean and maximum rates for the three locations are not significantly different (both one-way anovas). We have added Figure 7—figure supplement 2 showing the mean and maximum in field firing rates for LS cells in each location, and have added the following lines to the text:

“It does not appear that the higher cross correlations for cells on the choice side of the track were due to higher firing rates of cells proximal to reward, as there was no significant difference between the mean and maximum firing rates of LS place cells in all three locations (see Figure 7—figure supplement 2).”

The addition of Figure 3—figure supplement 2 also gives a clearer picture of firing rates across LS place cells, and we have added a histogram to this figure as well to represent all firing rates of LS cells within place fields.

4) Given the importance of LS place fields to this study, reviewers would like to see more than four examples. Reviewers suggested a Supplemental Figure that presents a large number of LS place fields, selected in an unbiased way.

Figure 3—figure supplement 2A was added which shows spatial firing rate heat maps of 30 randomly chosen LS cells.

5) Reviewers would also like to see examples of spike cross-correlations supporting the data in Figure 7.

Figure 7—figure supplement 1 was added which includes all cross correlations for data in Figure 7, as well as data for shuffled spike trains.

6) The authors note in the introduction that the hippocampal encoding of goal locations has been characterized, but they do not cite a highly-relevant study: Dupret et al., 2010. These experiments showed that the hippocampal over-representation of goal is not universal, but instead depends on the cognitive demands of the task. It is important to know, therefore, whether goal locations are over-represented in the present study. One straightforward way to address this would be to re-analyze the data from panels 6E-F. If the probability of a field were *per unit distance*, results from the different track segments would be directly comparable. This would reveal whether hippocampal place cells themselves are clustered near reward in this task, and to what degree LS neurons might amplify that clustering.

Great suggestion. We ran the data as suggested and added this data as panels on Figure 6. We found, to our positive surprise, that the LS, steadily increased the number of place fields as the animal approached the reward on the final arm. The hippocampus, in the final 60cm of award approach, appeared to drastically increase the number of place fields as well. However, this was offset somewhat by the large increase in fields 60-80cm from reward, which was mirrored by a large increase 200-220cm from reward. Both of these locations are proximal or overlapping the forced/choice point in the maze.

To explain this, we added the following text in the Results section of the paper:

“We also computed the probability of finding a place field as a function of distance from reward (Figure 6H-I). In the hippocampus, there was an increase in the probability of a spatial firing field in the last 60cm of reward approach (Figure 6H). However, the largest peeks in HPC place field probability were around the forced and choice points of the maze, approximately 200-220cm and 80-60cm away from reward, respectively. In the LS, the entire forced arms were highly overrepresented, and the probability of a place field also increased upon reward approach.”

We also added the following in the Discussion section:

“In the present task, the probability of finding a HPC place field increases as the animal approaches final goal location (Figure 6H). However, the representation of the goal is decreased relative to locations proximate to the forced and choice points of the maze. Past work has found that the HPC uniquely over-represents salient or goal locations on tasks demanding increased spatial memory (Dupret et al., 2010). It is possible, therefore, since the important spatial memory components of this task occurred at the forced and choice points, that these locations came to be even more over-represented in the HPC that the goal locations.”

7) The plot in Figure 7B shows an intriguing correlation. This quantification only shows the effect is present when averaging across all neurons, however. Does every neuron show such a correlation, or just a subpopulation? This information could help to corroborate or reject the authors' models. Related to the previous point, it would also provide an estimate of what fraction of hippocampal neurons might be specialized for encoding reward.

To answer this question, we shuffled the spike trains of all the HPC and LS pairs on the forced arm (Figure 7—figure supplement 1) and found an average correlation of 7.71e-05, with a 95% confidence interval of [-8.89e-05, 2.43e04]. Out of 36 unit pairs on the forced arm, the average of 26 of these pairs (72%) fell above the 95% confidence interval for shuffled data. Therefore, it appears that about 72% of pairs at the choice arm have a significantly higher correlation than would be expected by chance (and this number would clearly be higher if using a one-sided measurement). As pairing of neurons is not exact, I would also expect an even higher number of significant correlations with additional pairing options. Because the neurons pairs in the choice arm set are, by definition, proximal to reward, this finding appears to suggest that by virtue of being reward proximate the vast majority of HPC neurons are reward-encoding.

8) The recording location in the caudodorsal LS is very close to the septohippocampal nucleus (SHN) based on Paxinos and Watson's stereotaxic atlas, and indeed, at least one of the recording sites in Figure 1aAappears to be in the SHN. I am unaware of any literature quantifying place representation in the SHN, but given the controversy regarding whether LS neurons have place fields, it is important to determine whether the place-selective units in the current study are more likely to be LS or SHN neurons. Do the authors observe a correlation between medial/lateral (or other stereotaxic orientation) recording location and likelihood of observing place-specific firing? Perhaps in a supplemental analysis, the authors could replicate several of their core findings using only units recorded on the lateral-most tetrodes, which would be the least likely to be in the SHN.

Based on implant location and histology results, the number of tetrodes even potentially in the SHN is extremely small (<5), and we have recording data from over 100 tetrodes. Thus, it is impossible for tetrode placement in the SHN to explain the presence of place cells.

Additionally, based on the atlas, we are not even confident that these aforementioned 5 tetrodes are actually in the SHN versus the LS.

Additionally, previous work has found LS place fields spread throughout the entirety of the LS, including the more lateral and ventral regions (see our response to 1D above) (Bezzi et al., 2002; Kita et al., 1995; Leutgeb and Mizumori, 2002; Monaco et al.,, 2019; Nishijo et al., 1997; Takamura et al., 2006; Zhou et al., 1999).

Some anatomists, including Swanson, who published the seminal works on LS anatomy and chemoarchitecture , also consider that the SHN may be a part of the LS, and that the projections to and from the SHN (including the organization of these projects) are not markedly different from those seen in the remainder of the LS (Risold and Swanson 1997a,b). They also state “Furthermore, additional evidence now suggests that the septofimbrial nucleus, which lies caudomedially adjacent to the LS, as well as the tiny septohippocampal nucleus dorsomedially adjacent., may well constitute additional parts of what might be called the lateral septal complex, or at least should be considered with the LS.”

These considerations argue against redoing the analysis with a small subset of cells, particularly as including only the most lateral tetrodes would greatly diminish our statistical power.

9) In Figure 5, the authors quantify skewness and compare runs to reward vs. runs away from reward, reporting a significant difference for hippocampal fields, but not for LS fields. To facilitate interpretation of these results, reviewers would like to know if the skewness of HPC or LS neurons during runs to reward were different from zero (or different from a shuffle distribution). The same point was raised for runs away from reward.

A random distribution was obtained by shuffling firing rates within a firing field and then computing skew for the shuffled data. Because of a large variance in averages for shuffled data, averages for 500 shuffled distributions were computed. We added these shuffled averages as Figure 5—figure supplement 3. For hippocampal skew, the average skew for traveling to reward, traveling away from reward, and traveling away from reward for reward proximal cells all occurred less than 5% of the time in shuffled data. The skew result for skew traveling to reward for reward proximal cells did not fall outside 95% of shuffled data. These results have been added to Figure 5.

For the LS, the skew results for traveling away from reward, and reward proximal in both directions occurred less than 5% of the time in shuffled data. Traveling to reward occurred more than 5% of the time. These results have also been added to Figure 5.

We also used a one sample two-sided t-test to determine if any of the skew distributions had a mean significantly different than zero. The only significant result was for LS reward proximal cells while traveling away from reward (mean skew for LS away from reward is significantly different from zero, one sample two-sided t-test, t(58)=-2.0, p=0.05). This result has also been added to Figure 5.

10) Do LS cells have direction-specific place fields, and how does direction-selective firing (in HPC and LS neurons) impact skewness? Is the skewness driven entirely by uni-directional fields, or is it observed for both uni- and bi-directional fields?

The LS has both unidirectional and bidirectional place fields. A paragraph was added to Materials and methods section to describe how it was determined whether a field was uni or bi directional:

“Directionality was determined by computing place fields in both directions. If a unit had fields in both directions with centers separated less than 20cm, the field was considered bidirectional. For bidirectional place cells, skew was computed in both directions.”

We also added the following to the text to emphasize there are both uni and bi directional HPC and LS place fields:

“We wondered if place field location depending on direction of travel; for instance, if it was more likely to see a place field by a reward site after the site had been visited. To determine this, we split fields by direction, based on whether the animal was traveling to or from reward (if a field existed in both directions, we analyzed its parameters in both directions). This resulted in a total of 209 hippocampal place fields (115 towards reward and 94 away from reward, with, out of the total, 133 being unidirectional and 76 being bi directional) and 248 LS place fields (138 towards reward, 110 away from reward, with, out of the total, 177 unidirectional and 71 bidirectional. There was no significant difference of numbers of uni- or bi- directional HPC or LS cells, two-tailed two sample ttest, t(461)=1.377, p>0.05).”

The average LS skew is not significantly different for uni and bi-directional fields directional (two-tailed two sample t-test, t(246)=-0.19, p=0.85) and there are a wide range of skew values for both uni and bi-directional LS place fields. We also computed the values for uni and bi directional place cells based on direction of travel, and there was also no significant difference across the values (one way anova, F(3,244) = 1.05, p>0.05). We have added Figure 5— supplemental figure 4 to display these results.

We completed analogous analysis for HPC place cells and found analogous results: the mean skew was not significantly different between uni- and bi- directional place cells (two-tailed two sample t-test, t(207)=-0.59, p=0.55), nor did directionality matter for skew measured based on direction travelled relative to reward (one way anova, F(3,205) = 0.29, p>0.05). We have also added this data to Figure 5—figure supplement 4.

11) The datapoints in Figure 5 and Figure 5—figure supplement 2 don't seem to line up, and there was confusion about these figures. Is it correct that each dot represents a cell with its average place field peak at that location (relative to reward) and with that skewness? If that is correct, shouldn't the data in Figure 5F-G be a subset of the data in Figure 5—figure supplement 2? However, those points aren't the same. The difference should be made clear in the text or in the figure legend.

We thank the reviewers for catching this slip, which is also a result of the computational error reported above (Figure 5—figure supplement 1 is correct). After re-running the data, although there is a trend in the direction seen in originally in Figure 5F, it is not significant and the Figure 5F was moved to the supplement (now Figure 5—figure supplement 2). However, with this correction made, the result in Figure 5E is more striking than previously.

The finding has been discussed in the text as follows:

“We observed that the distribution of place fields in the lateral septum was more biased towards the rewarded locations of the maze than the distribution of place fields in CA1 (Figure 6), and that, unlike HPC place fields, LS place fields tended to skew towards reward direction regardless of the direction of travel, particularly when close to rewarded locations (Figure 5).”

[Editors’ note: further revisions were suggested prior to acceptance, as described below.]

Essential revisions:

1) It is still possible that the higher cross-correlations over many tens of milliseconds between HPC and LS cells with place fields in the choice side (Figure 7) can be secondary to a higher number of place fields in both regions in this part of the maze (Figure 6). This might be addressed, for instance, by comparing cross-correlations for forced, middle and choice sides for subsampled spike trains selected to ensure a matched degree of overlap of place fields in HPC and LS. However, the paper would still be interesting and worthy of publication even if it turns out that cross-correlations are due to concentration of Hip and LS place fields on the choice side close to rewards.

We understand and appreciate this concern. We first decided to determine if there was a difference in the average distance between LS and HPC place field pairs for the forced side, central stem, and choice side. When we completed this analysis, we found that there was no significant difference between the average distance between LS and HPC pairs in the forced arm versus choice arm, so the proximity of HPC-LS pairs did not account for the difference in cross correlation values for forced vs. choice sides. There was a small (3cm) but significant difference in distances when comparing the choice side to the middle stem.

We then followed the suggestion to subsample and found that eliminating the very closest pairs (pairs that had centers within 3cm of each other) was more than sufficient to result in an insignificant difference between forced, choice, and middle pair distances. However, the average cross correlation for pairs on the choice side was still significantly higher than the average in either the forced side or middle stem.

Therefore, it does not appear that differences in cross correlations are due to differences in the concentration of place fields in the forced, middle, and choice sides. We have added this to the text and created Figure 7—figure supplement 2.

2) Figure 5 seems to be consistent with the lack of influence of hippocampal inputs on the skewness of LS place fields. The latter are similarly skewed at different locations. However, HPC place fields change their directional skewness depending on the proximity of the reward. It would be helpful to integrate this finding in the models proposed.

The only difference in HPC skewness direction based on proximity occurs while traveling away from reward, where, looking at all HPC cells, the average skewness is negative, but for reward proximal cells, the average skewness is positive. However, the values for skew are not significantly different from one another (double sided t test t(131) = 1.62, p>0.05). We have added this clarification into the text for Figure 5: Note that while HPC skew away from reward for reward proximal cells appears to have a different direction than for all HPC cells when traveling away from reward, the two means are not significantly different (two sample two sided t-test, t(131)=1.62, p>0.05).